# Statistical Multicriteria Benchmarking via the GSD-Front

**Christoph Jansen**[1,*]
c.jansen@lancaster.ac.uk

**Georg Schollmeyer**[2,*]
georg.schollmeyer@stat.uni-muenchen.de

**Julian Rodemann**[2,*]
julian@stat.uni-muenchen.de

**Hannah Blocher**[2,*]
hannah.blocher@stat.uni-muenchen.de

**Thomas Augustin**[2]
thomas.augustin@stat.uni-muenchen.de

[1]School of Computing & Communications
Lancaster University Leipzig
Leipzig, Germany

[2]Department of Statistics
Ludwig-Maximilians-Universität München
Munich, Germany

## Abstract

Given the vast number of classifiers that have been (and continue to be) proposed, reliable methods for comparing them are becoming increasingly important. The desire for reliability is broken down into three main aspects: (1) Comparisons should allow for different quality metrics simultaneously. (2) Comparisons should take into account the statistical uncertainty induced by the choice of benchmark suite. (3) The robustness of the comparisons under small deviations in the underlying assumptions should be verifiable. To address (1), we propose to compare classifiers using a generalized stochastic dominance ordering (GSD) and present the GSD-front as an information-efficient alternative to the classical Pareto-front. For (2), we propose a consistent statistical estimator for the GSD-front and construct a statistical test for whether a (potentially new) classifier lies in the GSD-front of a set of state-of-the-art classifiers. For (3), we relax our proposed test using techniques from robust statistics and imprecise probabilities. We illustrate our concepts on the benchmark suite PMLB and on the platform OpenML.

## 1 Introduction

The comparison of classifiers in machine learning is usually carried out using *quality metrics* $\phi : \mathcal{C} \times \mathcal{D} \to [0, 1]$, i.e., bounded functions assigning a real number to every pair $(C, D)$ of classifier and data set from a suitable domain $\mathcal{C} \times \mathcal{D}$, where, by construction, higher numbers indicate better quality. However, in many applications, the choice of a unique quality metric used for the comparison is not self-evident. Instead, competing quality metrics are available, each of which can be well-motivated but may lead to a different ranking of the analyzed classifiers. One attempt to safeguard against this effect is to use *multidimensional quality metrics*: instead of a single metric, one chooses a set of metrics $\Phi := (\phi_1, \ldots, \phi_n) : \mathcal{C} \times \mathcal{D} \to [0, 1]^n$ that – taken together – provide a balanced

---

*marks equal contribution.

38th Conference on Neural Information Processing Systems (NeurIPS 2024).

foundation for assessing the quality of classifiers. Generally, we distinguish two (related but) different motivations for choosing multidimensional quality metrics:

**Performance is a latent construct:** The application at hand suggests a very clear evaluation concept, which, however, is too complex to be expressed in terms of a single metric. In this case, the *latent* construct to evaluate is operationalized with a set of quality metrics (that serve as an approximation). For example, the latent construct of *robust accuracy* can be operationalized by taking together the following three quality metrics: *Accuracy* of a classifier (i.e., the proportion of correctly predicted labels), and *robustness* of this proportion under weak perturbations of the data in either the features or the target variable. This will be exemplified in Section 5.2 using the PMLB benchmark suite.

**Quality is a multidimensional concept:** Even if the application at hand suggests evaluation criteria that can be perfectly expressed using quality metrics, it can still be desirable to compare the classifiers under consideration in terms of various contentual dimensions. For example, one can be interested in how well a classifier performs in the trade-off between *accuracy* and *computation time* in the training and the test phase: Clearly distinguishable contentual dimensions are included and the analysis aims at investigating how the different classifiers under consideration trade-off between these dimensions. This will be exemplified in Section 5.1 using one of OpenML's benchmark suites.

Regardless of the motivation for considering multidimensional quality metrics, their interpretative advantage naturally comes at a price: Without further assumptions, classifiers will often be incomparable, as the quality metrics in the different dimensions contradict each other in their ranking.[2] Already on one data set, a multidimensional quality metric only induces a (natural yet potentially incomplete) *preorder*: a classifier is rated at least as good as a competitor if (and only if) it receives at least the same metric value in each dimension. The problem of incomparability becomes even more severe for multiple data sets (as considered here). In this case, one of the following analysis paths is often chosen: (I) An expected *weighted sum* (for example, weighted by importance) of the individual quality metrics is considered and the problem is then analyzed on this new pooled quantity.[3] (II) The problem is analyzed based on the *Pareto-front* $\mathrm{par}(\Phi)$, i.e., the set of all classifiers that are not component-wise (strictly) dominated by any competitor, whose definition followed by an illustrative example are included for reference.

**Definition 1.** *Let $\tilde{\mathcal{D}} \subseteq \mathcal{D}$ be some set of data sets. The $\tilde{\mathcal{D}}$-**Pareto front** $\mathrm{par}(\Phi, \tilde{\mathcal{D}})$ of $\Phi$ is given by*

$$\left\{ C \in \mathcal{C} \, | \, \nexists C' \in \mathcal{C} \; \forall D \in \tilde{\mathcal{D}} : \; \Phi(C', D) \succ \Phi(C, D) \right\},$$

*where $\succ$ is the strict part of the component-wise $\geq$-relation on $\mathbb{R}^n$. Set $\mathrm{par}(\Phi) := \mathrm{par}(\Phi, \mathcal{D})$.*

**Example 1.** *Consider the following schematic example of three classifiers $\mathcal{C} = \{C_1, C_2, C_3\}$ evaluated for a fictitious population of four data sets $\mathcal{D} = \{D_1, D_2, D_3, D_4\}$. Every entry gives the two-dimensional evaluation $\Phi(C, D)$ of a classifier on a data set w.r.t. predictive accuracy and the computation time for training in three ordinal categories `fast`, `medium` and `slow`.*

| classifier \ data set | $D_1$ | $D_2$ | $D_3$ | $D_4$ |
|---|---|---|---|---|
| $C_1$ | $(0.7, \texttt{slow})$ | $(0.8, \texttt{medium})$ | $(0.9, \texttt{fast})$ | $(0.95, \texttt{slow})$ |
| $C_2$ | $(0.75, \texttt{slow})$ | $(0.85, \texttt{fast})$ | $(0.91, \texttt{fast})$ | $(0.96, \texttt{slow})$ |
| $C_3$ | $(0.99, \texttt{slow})$ | $(0.91, \texttt{fast})$ | $(0.85, \texttt{fast})$ | $(0.75, \texttt{slow})$ |

*Here, it holds that $\Phi(C_2, D_i) \succ \Phi(C_1, D_i)$ for all $i = 1, 2, 3, 4$, i.e., $C_1$ is component-wise (strictly) dominated by $C_2$. Classifiers $C_2$ and $C_3$ are not component-wise (strictly) dominated. Thus, the Pareto-front is given by $\mathrm{par}(\Phi) = \{C_2, C_3\}$.*

Both approaches are extreme in a certain sense: (I) reduces the multidimensional information structure of the problem to one single real-valued score. Any selection of classifier based on this score will heavily depend on the choice of the weights in the sum score and, therefore, becomes dubious once

---

[2]This effect is usually more pronounced under the second motivation: Whereas in the first case the different metrics attempt to formalize the same latent construct, here different quality dimensions are actually to be covered. E.g., an improvement in accuracy may often be accompanied by a deterioration in computation time.

[3]There, one interprets the data sets as realizations of a random variable $T : \Omega \to \mathcal{D}$ on some probability space $(\Omega, \mathcal{S}, \pi)$, chooses weights $w_1, \ldots, w_n \in \mathbb{R}_+$ and assigns each $C \in \mathcal{C}$ the value $\sum_{i=1}^n w_i \mathbb{E}_\pi(\phi_i(C, T))$.

this choice is not perfectly justified. This seems even more severe for problems where some of the involved quality metrics might only allow for an ordinal interpretation, e.g., feature sparseness as a proxy for interpretability [75], risk levels in the EU AI act [50] or other regulatory frameworks [73], robustness (see experiments in Section 5.2) or runtime levels (Section 5.1). Opposed to this, (II) seems to be very conservative: By considering classifiers that are in the Pareto-front par$(\Phi)$, one (potentially) completely ignores both information encoded in the cardinally interpretable dimensions and information about the distribution of the data sets. As a trade-off between these two extremes, which utilizes the complete available information but avoids the choice of weights, it has recently been proposed to compare classifiers under multidimensional quality metrics using *generalized stochastic dominance (GSD)* [45]. The rough idea of this approach is to first embed the range of the multivariate performance measure in a special type of relational structure, a so-called *preference system*, which then allows for also formalizing the entire information originating from the cardinal dimensions of the quality metric. A classifier is then judged at least as good as a competitor (similar to classic stochastic dominance), if its expected utility is at least as high with respect to every utility function representing (both the ordinal and the cardinal parts of) the preference system (also see Definition 5). Although GSD also induces only a preorder, the set of not strictly dominated classifiers will generally be considerably smaller than under the classical Pareto analysis. Furthermore, it avoids potentially difficult to justify assumptions about the weighting of the different quality metrics. Therefore, working with the GSD-front, as introduced below, will prove to be a very promising analysis option; it combines the advantages of the conservative Pareto analysis with those of the liberal comparison of weighted sums.

## 1.1 Our contribution

**GSD-Front:** We introduce the concept of the GSD-front (see, after some preparations, Definition 6) and characterize it in Theorem 2 as more discriminative than the Pareto-front. In this sense, the GSD-front is an information-efficient way to handle the multiplicity/implicit multidimensionality of quality criteria, powerfully exploiting their ordinal and quantitative components.

**Proper handling of statistical uncertainty; estimating and testing:** Since typically the available data sets are just a sample of the corresponding universe, empirical counterparts of the major concepts are needed to do justice to the underlying statistical uncertainty. In particular, we give a sound inference framework: Firstly, we propose a set-valued estimator for the GSD-front and provide sufficient conditions for its consistency (see Theorem 1 and Remark 3). Secondly, we develop static and dynamic statistical permutation-tests if a classifier is in the GSD-front and prove their level-$\alpha-$ validity and their consistency (see Theorem 3).

**Robustification:** Additionally, we recognize the fact that the underlying assumption of identically and independently distributed *(i.i.d.)* sampling is questionable in many benchmarking studies. Thus, in Section 4.2 we quantify how robust the test decisions are under such deviations.

**Experiments with benchmark suites and implementation:** We illustrate the concepts and corroborate their relevance with experiments run over two benchmark suites (PMLB and OpenML, see Section 5), based on an implementation that is freely available and easily adaptable to comparable problems.[4]. We consider experiments with *mixed-scaled* (ordinal and cardinal) multidimensional quality metrics, also incorporating (potentially) ordinal criteria.

## 1.2 Related work

*Benchmarks* are the foundations of applied machine learning research [27, 90, 78, 65, 91]. Specifically, benchmarking classifiers over multiple data sets is a much-studied problem in machine learning, as it enables practitioners to make informed choices about which methods to consider for a given data set. Furthermore, also proposals for novel classifiers must often first demonstrate their potential for improvement in benchmark studies. Examples include [58, 40, 31, 57, 12]. In recent years, in recognition of the fact that the benchmark suite under consideration is only a sample of data sets, especially focusing on *statistically significant* differences between classifiers has received great interest (see, e.g., [24, 35, 34, 19, 45] or, e.g., [9, 22, 8] for Bayesian variants). An R implementation of some of these tests is described in [15], whereas use-cases in the context of time series and

---

[4]Implementations of all methods and scripts to reproduce the experiments:`https://github.com/hannahblo/Statistical-Multicriteria-Benchmarking-via-the-GSD-Front`.

neural networks for regression are discussed in [44, 36]. The diversity and the associated problem of selecting quality metrics (e.g., [51]) is currently attracting a great deal of interest (e.g., [89]). Consequently, finding ways for comparing classifiers in terms of *multidimensional quality metrics* is intensively studied, ranging from multidimensional interpretability measures (e.g., [59]) over classical Pareto-analyses (e.g., [31]) to embeddings in the theory of data depth (e.g., [13, 71]). While utilizing variants of *stochastic dominance* in statistics is quite common (e.g., [56, 61, 6, 76, 67]), the same seems not to hold for machine learning. Exceptions include [23] in an optimization context, [47, 48], who investigate special types of stochastic orders, and [45], utilizing already GSD-relations for classifier comparisons without the GSD-front. Finally, relying on imprecise probabilities (e.g., [85, 86, 3]) to robustify statistical hypotheses follows the tradition of [66, 42, 41, 2], see also, e.g., [5, 25, 4, 60, 48]. For application to Bayesian networks, see, e.g, [55, 14, 54], and [81, 69, 18, 80, 1, 70, 52, 30, 16, 17], among others, for robustified machine learning in this spirit.

## 2   Decision-theoretic preliminaries

The relevant basic concepts in order theory are collected in Appendix A.1. Based on these we can make the following definition, originating from the decision-theoretic context discussed in [46].

**Definition 2.** *Let $A$ be a non-empty set, $R_1 \subseteq A \times A$ a preorder on $A$, and $R_2 \subseteq R_1 \times R_1$ a preorder on $R_1$. The triplet $\mathcal{A} = [A, R_1, R_2]$ is then called a **preference system** on $A$. The preference system $\mathcal{A}' = [A', R_1', R_2']$ is called **subsystem** of $\mathcal{A}$ if $A' \subseteq A$, $R_1' \subseteq R_1$, and $R_2' \subseteq R_2$.*

In our context, $R_1$ formalizes the ordinal information, i.e., the information about the ranking of the objects in $A$, whereas $R_2$ describes the cardinal information, i.e., the information about the intensity of certain rankings. To ensure that $R_1$ and $R_2$ are compatible, we use a consistency criterion relying on the idea of simultaneous representability of both relations. For this, for a preorder $R$, we denote by $I_R$ its indifference and by $P_R$ its strict part (see A.1).

**Definition 3.** *The preference system $\mathcal{A} = [A, R_1, R_2]$ is **consistent** if there exists a **representation** $u : A \to \mathbb{R}$ such that for all $a, b, c, d \in A$ we have:*

  *i)  $(a, b) \in R_1 \Rightarrow u(a) \geq u(b)$ with equality iff $(a, b) \in I_{R_1}$*

  *ii)  $((a, b), (c, d)) \in R_2 \Rightarrow u(a) - u(b) \geq u(c) - u(d)$ with equality iff $((a, b), (c, d)) \in I_{R_2}$*

*The set of all representations of $\mathcal{A}$ is denoted by $\mathcal{U}_\mathcal{A}$.*

Finally, we need to recall the concept of *generalized stochastic dominance (GSD)* (see, e.g., [48]), which is crucial for the concepts presented in this paper: For a probability space $(\Omega, \mathcal{S}, \pi)$ and a consistent preference system $\mathcal{A}$, we define by $\mathcal{F}_{(\mathcal{A}, \pi)}$ the set of all $X \in A^\Omega$ such that $u \circ X \in \mathcal{L}^1(\Omega, \mathcal{S}, \pi)$ for all $u \in \mathcal{U}_\mathcal{A}$. We then can define the GSD-preorder on $\mathcal{F}_{(\mathcal{A}, \pi)}$ as follows.

**Definition 4.** *Let $\mathcal{A} = [A, R_1, R_2]$ be consistent. For $X, Y \in \mathcal{F}_{(\mathcal{A}, \pi)}$, say $X$ $(\mathcal{A}, \pi)$-**dominates** $Y$ if $\mathbb{E}_\pi(u \circ X) \geq \mathbb{E}_\pi(u \circ Y)$ for all $u \in \mathcal{U}_\mathcal{A}$. Denote the induced **GSD-preorder** on $\mathcal{F}_{(\mathcal{A}, \pi)}$ by $R_{(\mathcal{A}, \pi)}$.*

## 3   GSD for classifier comparison

We return to the initial problem: Assume we are given a finite set $\mathcal{C}$ of classifiers, an arbitrary set $\mathcal{D}$ of data sets and $n$ quality metrics $\phi_1, \ldots, \phi_n : \mathcal{C} \times \mathcal{D} \to [0, 1]$, combined to the multidimensional quality metric $\Phi := (\phi_1, \ldots, \phi_n) : \mathcal{C} \times \mathcal{D} \to [0, 1]^n$. As we also want to allow ordinal quality metrics, we assume that, for $0 \leq z \leq n$, the metrics $\phi_1, \ldots, \phi_z$ are of cardinal scale (differences may be interpreted), while the remaining ones are purely ordinal (differences are meaningless apart from sign). We embed the range $\Phi(\mathcal{C} \times \mathcal{D})$ of $\Phi$ in the following preference system:

$$\mathcal{P} = [[0, 1]^n, R_1^*, R_2^*] \text{, where} \tag{1}$$

$$R_1^* = \left\{ (x, y) : x_j \geq y_j \ \forall j \leq n \right\}, \text{ and } R_2^* = \left\{ ((x, y), (x', y')) : \begin{array}{l} x_j - y_j \geq x_j' - y_j' \ \forall j \leq z \\ x_j \geq x_j' \geq y_j' \geq y_j \ \forall j > z \end{array} \right\}.$$

$R_1^*$ is the usual component-wise $\geq$-relation. For $R_2^*$, one pair of consequences is preferred to another if, in the ordinal dimensions, the exchange associated with the first pair is not a deterioration to the

exchange associated with the second pair and, in addition, there is component-wise dominance of the differences of the cardinal dimensions. In order to transfer the GSD-relation from Definition 4 to the case of comparing classifiers under multidimensional performance metrics, we interpret the data sets in $\mathcal{D}$ as realizations of a random variable $T : \Omega \to \mathcal{D}$ on some probability space $(\Omega, \mathcal{S}, \pi)$. We then associate each classifier $C \in \mathcal{C}$ with the random variable $\Phi_C := \Phi(C, T(\cdot))$ on $\Omega$ and compare classifiers by comparing the associated random variables by means of GSD.

**Definition 5.** *Denote by $\mathcal{P}_\Phi$ the preference system obtained by restricting $\mathcal{P}$ to $\Phi(\mathcal{C} \times \mathcal{D})$. Further, let $\mathcal{C}$ be such that $\{\Phi_C : C \in \mathcal{C}\} \subseteq \mathcal{F}_{(\mathcal{P}_\Phi, \pi)}$. For $C, C' \in \mathcal{C}$, say that $C$ **dominates** $C'$, abbreviated with $C \succsim C'$, whenever $(\Phi_C, \Phi_{C'}) \in R_{(\mathcal{P}_\Phi, \pi)}$.*

In the application situation, instead of the true GSD-order $\succsim$ among classifiers, we will often have to get along with its *empirical analogue*, i.e., the GSD-relation where a sample of data sets is treated like the underlying population and the true probability measure is replaced by the corresponding empirical ones. More precisely, we assume that we have sampled *i.i.d.* copies $T_1, \ldots, T_s$ of $T$ and then define the set $Z_s := \{\Phi(C, T_i) : i \leq s \wedge C \in \mathcal{C}\}$, of (random) observations under the different classifiers. We then use $\mathcal{W}$ to denote the (random) subsystem of $\mathcal{P}$ that arises when $\mathcal{P}$ is restricted to the (random) set $Z_s$. For $C, C' \in \mathcal{C}$ we define the random variable

$$d_s(C, C') := \inf_{u \in \mathcal{U}_\mathcal{W}} \sum_{z \in Z_s} u(z)(\hat{\pi}_C(\{z\}) - \hat{\pi}_{C'}(\{z\})),$$

where, for $M \subseteq [0,1]^n$, we set $\hat{\pi}_C(M) := \frac{1}{s}|\{i : i \leq s \wedge \Phi(C, T_i) \in M\}|$. For a concrete sample associated to $\omega_0 \in \Omega$, we then say that $C$ *empirically GSD-dominates* $C'$, if $d_s(C, C')(\omega_0) \geq 0$. Intuitively, $d_s$ can thus be used to check whether the classifier $C$ empirically dominates the classifier $C'$ with respect to GSD in the samples at hand (i.e., in the benchmark suite under investigation).

Based on these concepts, we can now define the sets of (empirically) GSD-undominated classifiers.

**Definition 6.** *Let $\mathcal{C}$ be such that $\{\Phi_C : C \in \mathcal{C}\} \subseteq \mathcal{F}_{(\mathcal{P}_\Phi, \pi)}$. Let denote $T_1, \ldots, T_s$ i.i.d. copies of $T$.*

  *i) The **GSD-front** is the set*

$$gsd(\mathcal{C}) := \{C \in \mathcal{C} : \nexists C' \in \mathcal{C} \text{ s.t. } C' \succ C\},$$

  *where $\succ$ denotes the strict part of $\succsim$.*

  *ii) Let $\varepsilon \in [0,1]$. The $\varepsilon$-**empirical GSD-front** is the (random) subset of $\mathcal{C}$ defined by*

$$egsd_s^\varepsilon(\mathcal{C}) = \left\{ C : \nexists C' \in \mathcal{C} \text{ s.t. } \begin{array}{l} d_s(C', C) \geq -\varepsilon \\ d_s(C, C') < 0 \end{array} \right\}.$$

**Remark 1.** *$egsd_s^0(\mathcal{C})$ is always non-empty. In contrast, $egsd_s^\varepsilon(\mathcal{C})$ may very well be empty if $\varepsilon > 0$. Note that choosing values of $\varepsilon > 0$ is intended to make $egsd_s^\varepsilon(\mathcal{C})$ less prone to sampling noise.*

**Remark 2.** *Some words on the semantics of the GSD-front: From a decision-theoretic point of view, classifier $C$ strictly GSD-dominates classifier $C'$ iff $C$ has at least as high expected utility as $C$ regarding any compatible utility representation of all the metrics considered, and stricly higher for at least one such utility. The GSD-front then simply collects all classifiers from $\mathcal{C}$ which are not strictly GSD-dominated by any competitor, i.e., which potentially can be optimal in expectation.*

**Example 2.** *Consider again the situation of Example 1 and recall that $par(\Phi) = \{C_2, C_3\}$ leaves $C_2$ and $C_3$ incomparable. However, if considering only the distribution of the (multivariate) performance of the classifiers (while assuming a uniform distribution over $\mathcal{D}$), $C_3$ is clearly dominating $C_2$ w.r.t. GSD: Matching dataset $D_i$ with dataset $D_{5-i}$ creates a (strict) pointwise dominance of $C_3$ over $C_2$ (where the strict dominance is due to $D_1$ and $D_4$). Thus, $gsd(\mathcal{C}) = \{C_3\} \subsetneq par(\Phi) = \{C_2, C_3\}$.*

The following two theorems show that the $\varepsilon$-empirical GSD-front fulfills two very natural requirements: First, under some regularity conditions, it is a consistent statistical estimator for the true GSD-front (Theorem 1). This is important because in practical benchmarking we almost never have access to the GSD-front of the whole population, i.e., the benchmarking results on all possible datasets from a specific problem class $\mathcal{D}$. Second, it is ensured that neither the $\varepsilon$-empirical nor the true GSD-front can ever become larger than the respective Pareto-front, irrespective of the choice of $\varepsilon$ (Theorem 2). This is important as it guarantees our analysis does never conflict with, but is potentially more information-efficient than a Pareto-type analysis. Proofs are given in B.1 and B.2.

**Theorem 1.** *Denote by $\mathcal{I}_\Phi$ the set of all sets $\{a : u(a) \geq c\}$, where $c \in [0,1]$ and $u \in \mathcal{U}_{\mathcal{P}_\Phi}$. Assume that $\succsim$ is antisymmetric. If the VC-dimension[5] of $\mathcal{I}_\Phi$ is finite and if $\varepsilon : \mathbb{N} \to [0,1]$ converges to 0 with rate at most $\Theta(1/\sqrt[4]{s})$, then $(\mathrm{egsd}_s^{\varepsilon(s)}(\mathcal{C}))_{s\in\mathbb{N}}$ is a consistent statistical estimator, i.e.,*

$$\pi\Big(\Big\{\omega \in \Omega : \lim_{s\to\infty} \mathrm{egsd}_s^{\varepsilon(s)}(\mathcal{C}) = \mathrm{gsd}(\mathcal{C})\Big\}\Big) = 1,$$

*where set convergence is defined via the trivial metric.*

**Remark 3.** *The assumption of a finite VC dimension is only necessary to ensure that the $\varepsilon$-empirical GSD front does not become too large. In particular, the following does hold **without** this assumption:*

$$\pi\Big(\Big\{\omega \in \Omega : \lim_{s\to\infty} egsd_s^{\varepsilon(s)}(\mathcal{C}) \supseteq gsd(\mathcal{C})\Big\}\Big) = 1.$$

*Thus, the $\varepsilon$-empirical GSD-front almost surely converges to a superset of the true GSD-front.*

**Theorem 2.** *Assume $\mathcal{C}$ with $\{\Phi_C : C \in \mathcal{C}\} \subseteq \mathcal{F}_{(\mathcal{P},\pi)}$. Let further denote $T_1, \ldots, T_s$ i.i.d. copies of $T$ and let $\varepsilon_1 \leq \varepsilon_2 \in [0,1]$. It then holds that i) $\mathrm{gsd}(\mathcal{C}) \subseteq \mathrm{par}(\Phi)$. Moreover, it holds that ii) $\mathrm{egsd}_s^{\varepsilon_2}(\mathcal{C}) \subseteq \mathrm{egsd}_s^{\varepsilon_1}(\mathcal{C}) \subseteq \mathrm{par}(\Phi, \{T_1, \ldots, T_s\})$.*

## 4 Statistical testing

We saw the $\varepsilon$-empirical GSD-front can be a consistent statistical estimator and that both the empirical and the true GSD-front are compatible with the Pareto-front. We now address statistical testing.

### 4.1 A test for the GSD-front

From now on, we make the (technical) assumption that the order $\succsim$ among the classifiers from $\mathcal{C}$ is additionally *antisymmetric*, transforming it from a preorder into a partial order.[6] Equipped with this assumption, we want to address the question how to *statistically test* if a given classifier $C \in \mathcal{C}$ is an element of the true GSD-front $\mathrm{gsd}(\mathcal{C})$. To achieve this, we formulate the question of actual interest as the alternative hypothesis of the test, i.e., we obtain the hypothesis pair:

$$H_0 : C \notin \mathrm{gsd}(\mathcal{C}) \quad \textbf{vs.} \quad H_1 : C \in \mathrm{gsd}(\mathcal{C})$$

A possible motivation for developing tests on the hypothesis pair $(H_0, \neg H_0)$ is the following: One would like to compare the quality of a newly developed classifier $C$ for a problem class $\mathcal{D}$ with the classifiers in $\mathcal{C} \setminus \{C\}$ that are considered state-of-the-art for this problem class, see application in Section 5.2. If a suitable statistical test would allow the above null hypothesis to be rejected, then one could draw the conclusion (subject to statistical uncertainty) that the new classifier $C$ on the problem class $\mathcal{D}$ could potentially improve the state-of-the-art. As first step, note that (under asymmetry) the null hypothesis $H_0$ can be equivalently rewritten as $H_0 : \exists C' \in \mathcal{C} \setminus \{C\} : C' \succsim C$. This reformulation makes obvious that $H_0$ is false if and only if *for every* $C' \in \mathcal{C} \setminus \{C\}$ the auxiliary hypothesis $H_0^{C'} : C' \succsim C$ is false. Statistical tests for hypothesis pairs of the form $(H_0^{C'}, \neg H_0^{C'})$ were proposed (in the context of statistical inequality analysis) in [48]: The authors there showed how exact statistical tests under *i.i.d.* sampling can be constructed by using a (non-parametric) permutation test based on a regularized version $d_s^\delta(C', C)$ of $d_s(C', C)$ as a test statistic. The strength of regularization of the test statistic is there controlled by a parameter $\delta \in [0,1]$, whose increase reduces the number of representation functions over which the infimum in the test statistic is formed, while equally attenuating all quality metrics.[7] Due to space limitations, we omit to recall an exact description of the testing scheme in the main text and instead refer to Appendix A.2.

The idea is then to replace the global test for $(H_0, \neg H_0)$ with $c := |\mathcal{C}| - 1$ tests of hypotheses $(H_0^{C'}, \neg H_0^{C'})$ and to reject the null hypothesis at significance level $\alpha$ if all tests reject their individual null hypotheses $H_0^{C'}$ at the same significance level $\alpha$. Call this the **static GSD-test**. Clearly, this test tends to be conservative, as it ignores potential correlations of the test statistics for different pairs of classifiers. Moreover, a slightly modified test in the context of the GSD-front is directly

---

[5]The VC-dimension of a set system $\mathcal{S}$ is the largest cardinality of a set $A$ with $2^A = \{A \cap S : S \in \mathcal{S}\}$.

[6]This is not very restrictive, it only assumes to consider classifiers that are not already equivalent w.r.t. GSD.

[7]In both applications in Section 5 the tests are based on the unregularized statistics $d_s^0(C', C)$, as the regularization performed in [48] aims at reaching a goal which is not primarily relevant for our paper (see A.2.5).

derivable: If one is rather interested in identifying the maximal subset $\mathcal{S}_{\max}$ of $\mathcal{C}$ for which $C$ significantly lies in the GSD-front, i.e., in testing $\tilde{H}_0^{\mathcal{S}} : C \notin \mathrm{gsd}(\mathcal{S})$ **vs.** $\tilde{H}_1^{\mathcal{S}} : C \in \mathrm{gsd}(\mathcal{S})$ for all $\mathcal{S} \subseteq \mathcal{C}$ with $C \in \mathcal{S}$ *simultaneously*, the following alternative test is a statistically valid level-$\alpha$ test: First, perform all individual tests for $(H_0^{C'}, \neg H_0^{C'})$ with level $\frac{\alpha}{c}$. Then identify $\mathcal{S}_{\max}$ as the set of all classifiers from $\mathcal{C}$ for which the individual hypotheses are rejected. The (random) alternative hypothesis $\tilde{H}_1^{\mathcal{S}_{\max}} : C \in \mathrm{gsd}(\mathcal{S}_{\max})$ is then statistically valid in the sense of being false only with a probability bounded by $\alpha$. Call this the **dynamic GSD-test**. We have the following theorem, demonstrating that the proposed tests are indeed reasonable statistical tests (see B.3 for the proof).

**Theorem 3.** *Let the assumptions of Theorem 1 hold. Then, both the static and dynamic GSD-test are valid level-$\alpha$ tests. Additionally, both tests are consistent in the sense that under the corresponding alternative hypothesis, i.e., $H_1 : C \in \mathrm{gsd}(\mathcal{C})$ resp. $\tilde{H}_1 : \exists \mathcal{S} \subseteq \mathcal{C} : C \in \mathcal{S}, |\mathcal{S}| \geq 2, C \in \mathrm{gsd}(\mathcal{S})$, the probability of rejecting the corresponding null hypothesis converges to 1 as $s \to \infty$.*

## 4.2 Checking robustness under non-i.i.d.-scenarios

We argue that meaningful benchmark studies should abstain from treating the sample of data sets in the suite as a *complete survey*. That is, benchmark analyses should aim at statements about a well-defined population and regard the benchmark suite as a non-degenerate sample thereof. A major practical problem in this context is that often little is known about the inclusion criteria for data sets or test problems in the respective benchmark suite (see, e.g., the discussions in [83, 53, 37]). For instance, the popular platform OpenML [82] allows users to upload benchmark results for machine learning models with varying hyperparameters, harming representativity, see Section 5.1 and Appendix C.1. The absence of methods to randomly sample from the set of all problems or data sets is identified as an unsolved issue in [57, Section 2]. This calls the common *i.i.d.* sampling assumption into question, which our (and most other) tests are based upon, and raises the issue as to what extent statistically significant results depend on this assumption. We now address precisely this question.

In [48] it was shown how the binary tests on the hypothesis pairs $(H_0^{C'}, \neg H_0^{C'})$ discussed in Section 4.1 can be checked for robustness against deviations from the underlying *i.i.d.*-assumption. The idea here is to deliberately perturb the empirical distributions of the performances for the different classifiers and to analyze the permutation test used under the most extreme yet compatible worst-case. The perturbation of the empirical distribution is carried out here using a $\gamma$-contamination model (see, e.g., [85, p. 147]), which is widely used in robust statistics. We now want to adapt a similar robustness check for the global hypothesis pair $(H_0, \neg H_0)$ discussed here. For this, suppose we have a sample $T_1, \ldots, T_s$ of data sets (i.e., the benchmark suite). We further assume that $k \leq s$ of these variables (where it is not known which ones) are not sampled *i.i.d.*, but come from an arbitrary distribution about which nothing else is known. We then know, for every fixed $C \in \mathcal{C}$, that its associated true empirical measure $\hat{\pi}_C^{true}$ based on the true (uncontaminated) sample would have to be contained in

$$\mathcal{M}_C = \left\{ (1 - \tfrac{k}{s})\hat{\pi}_C^{cont} + \tfrac{k}{s}\mu : \mu \text{ probability measure} \right\}, \qquad (2)$$

where $\hat{\pi}_C^{cont}$ denotes the empirical measure based on the contaminated sample $T_1, \ldots, T_s$. Note that $\mathcal{M}_C$ is by definition a $\gamma$-contamination model with central distribution $\hat{\pi}_C^{cont}$ and contamination degree $\gamma := \tfrac{k}{s}$. In this setting, [48] show that to ensure that their permutation tests used for hypothesis pairs $(H_0^{C'}, \neg H_0^{C'})$ only advise rejection of the null hypothesis if this is justifiable for any empirical distribution compatible with the contaminated sample, i.e., for every combination of measures $(\pi_1, \pi_2) \in \mathcal{M}_C \times \mathcal{M}_{C'}$, one has to compare the most pessimistic value of the test statistic for the concrete sample at hand with the most optimistic value of the test in each of the resamples. Moreover, they show that the (approximate) *observed p-values* for a concrete contaminated sample $T_1(\omega_0), \ldots, T_s(\omega_0) \in \mathcal{D}$ associated with $\omega_0 \in \Omega$ of this robustified test can be expressed by a function in the number of contaminations $k$, given by

$$f_{(C',C)}(k) := 1 - \tfrac{1}{N} \cdot \sum_{I \in \mathcal{I}_N} \mathbb{1}_{\left\{ d_I^\delta - d_s^\delta(C',C)(\omega_0) > \frac{2k}{(s-k)} \right\}},$$

where $N$ denotes the number of resamples, $\mathcal{I}_N$ is the corresponding set of resamples, and $d_I^\delta$ is the test statistic evaluated for the resample associated to $I$. Due to space limitations, we omit an exact description of the robustness check for the test on the hypothesis pairs $(H_0^{C'}, \neg H_0^{C'})$ as well as a derivation of the function $f_{(C',C)}$ in the main text and instead refer to Appendix A.3.

Similar as shown in Section 4.1, it is straightforward to calculate an (approximate) observed $p$-value for the static GSD-test for $(H_0, \neg H_0)$: We calculate the maximal observed $p$-value among all $C' \in \mathcal{C} \setminus \{C\}$, i.e. set $F_C(k) := \max\{f_{(C',C)}(k) : C' \in \mathcal{C} \setminus \{C\}\}$. The **robustified static GSD-test** for the degree of contamination $k$ can be carried out as follows: Calculate $F_C(k)$ and reject $H_0$ if $F_C(k) \leq \alpha$. This indeed gives us a valid level-$\alpha$-test for the desired global hypothesis $H_0 : C \notin \mathrm{gsd}(\mathcal{C})$ under the additional freedom that up to $k$ of the variables in the sample might be contaminated. Note, however, that also this test tends to be conservative as both performing the individual tests at level $\alpha$ as well as the adapted resampling scheme of the permutation test are worst-case analyses. Finally, also the **robustified dynamic GSD-test** can be obtained straightforwardly: Under up to $k$ contaminations, the (random) alternative hypothesis $\tilde{H}_1^{\mathcal{S}_{\max}} : C \in \mathrm{gsd}(\mathcal{S}_{\max})$ is statistically valid with level $\alpha$ if all individual robustified tests reject $H_0^{C'}$ at level $\frac{\alpha}{c}$, i.e., if $F_C(k) \leq \frac{\alpha}{c}$.

We end the section with a short comment on computation: The test statistics for the permutation test and the robustified variant can be calculated using linear programming. We are guided here by the linear programs proposed in [48, Propositions 4 and 5]. There are two computational bottlenecks in the actual evaluation: (1) the creation and storage of the constraint matrices of the linear programs and (2) the repeated need to solve large linear programs. An efficient, well-commented implementation that can be quickly transferred to similar applications is made available on GitHub (see Footnote 4).

## 5 Benchmarking experiments

We demonstrate our concepts on two well-established benchmark suites: OpenML [82, 11] and PMLB [64]. While for PMLB we compare classifiers w.r.t. the latent quality metric robust accuracy (see the first motivation in Section 1), for OpenML we use a multidimensional metric that includes accuracy and computation time as unidimensional metrics (see the second motivation in Section 1). The analysis of PMLB is kept short in the main text and detailed in Appendix C. Since the metrics in both applications are composed of one continuous and two (finitely) discrete metrics, we have (see B.4):

**Corollary 1.** *In both applications, the $\varepsilon$-empirical GSD-front is a consistent estimator for the true GSD-front (provided $\varepsilon$ is chosen as in Theorem 1).*

### 5.1 Experiments on OpenML

We select 80 binary classification datasets (according to criteria detailed in Appendix C.1) from OpenML [82] to compare the performance of *Support Vector Machine* (SVM) with *Random Forest* (RF), *Decision Tree* (CART), *Logistic Regression* (LR), *Generalized Linear Model with Elastic net* (GLMNet), *Extreme Gradient Boosting* (xGBoost), and *k-Nearest Neighbors* (kNN).[8] Our multidimensional quality metric is composed of *predictive accuracy*, *computation time on the test data*, and *computation time on the training data*. Since the computation time depends strongly on the used computing environment (e.g. number of cores or free memory), we discretize the time-related metrics and treat them as ordinal. Accuracy is not affected by this and is therefore treated as cardinal. For details, see Appendix C.1. To gain a purely descriptive impression, we computed the empirical GSD relation. For this, we calculated $d_{80}(C, C')$ for $C \neq C' \in \mathcal{C} := \{$SVM, RF, CART, LR, GLMNet, xGBoost, kNN$\}$ (see Hasse graph in Figure 2 in Appendix C.1). We see that CART (strictly) empirically GSD-dominates xGBoost, SVM, LR, and GLMNet. All other classifiers are pairwise incomparable. Three classifiers are not strictly empirically GSD-dominated by any other, namely RF, CART, and kNN. Thus, the $0$-empirical GSD-front is formed by these. While at first glance this result might seem rather unexpected, a closer look on the performance evaluations provided by OpenML indeed confirms the dominance structure found, see Appendix C.1 for details.

To move to reliable inferential statements that take into account the statistical uncertainty, we exemplarily test (at level $\alpha = 0.05$) if SVM significantly lies in the GSD-front of some subset of $\mathcal{C}$. As described in Section 4.1, we therefore perform six pairwise permutation tests for the hypothesis pairs $(H_0^{C'}, \neg H_0^{C'})$ (where $C := $ SVM and $C' \in \mathcal{C} \setminus \{$SVM$\}$) at level $\alpha$ in case of the **static GSD-test** or at level $\frac{\alpha}{6}$ in case of the **dynamic GSD-test**.[9] That is, we test six auxiliary null hypotheses each stating that SVM is GSD-dominated by kNN, xGBoost, RF, CART, LR, and GLMNet, respectively.

---

[8]For benchmarking deep learning classifiers or optimizers, we refer to future work discussed in Section 6.

[9]As explained in Footnote 7, we base the tests in Sections 5.1 and 5.2 on the unregularized $d_s^0(C', C)$.

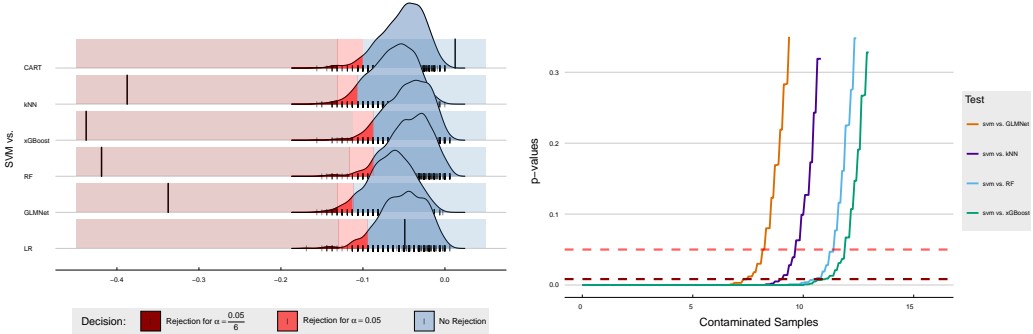

Figure 1: Left: Densities of resampled test statistics for pairwise permutation tests of SVM vs. six other classifiers on 80 datasets from OpenML. Big (small) vertical lines depict observed (resampled) test statistics. Rejection regions for the static (dynamic) GSD-test are highlighted red (dark red). Right: Effect of Contamination: $p$-values for pairwise tests of SVM versus GLMNet, kNN, RF and xGBoost. Red lines mark significance levels of $\alpha = 0.05$ (dark red: $\alpha = \frac{0.05}{6}$). Significance of SVM being in the GSD-front remains stable under contamination of up to 7 of 80 datasets.

The distributions of the test statistics are visualized on the left of Figure 1 (densities) and Figure 3 (CDFs) in C.1. They show that the pairwise tests of SVM versus kNN, xGBoost, RF, and GLMNet reject at level $\frac{\alpha}{6}$ and, thus, that SVM significantly (at level $\alpha$) lies in the GSD-front of the subset of $\mathcal{C}$ composed of SVM and these four classifiers. In other words, we conclude that SVM is significantly ($\alpha = 0.05$) not outperformed by kNN, xGBoost, RF, and GLMNet regarding all compatible utility representation of accuracy, training and test runtime. Finally, as discussed in Section 4.2, we turn to the third aspect of reliability (besides multiple criteria and statistical uncertainty): We analyze how robust this test decision is under contamination of the benchmark suite, i.e., deviations from *i.i.d.*. The results are visualized on the right of Figure 1. It can be seen that the tests at level $\frac{0.5}{6}$ of SVM against GLMNet, kNN, RF and xGBoost cease to be significant from a contamination of (approximately) 7, 8, 11, and 11 of 80 data sets, respectively. That is, the results on up to 7, 8, 11, and 11 datasets could be arbitrarily redistributed, while maintaining significance of rejection. Since the significance of the dynamic GSD-test's decision depends on all pairwise tests being significant at level $\frac{0.5}{6}$, we can conclude that SVM would still have been significantly in the GSD-front of {SVM, kNN, xGBoost, RF, GLMNet}, even if 7 out of 80 data sets had been contaminated. Summing up, our proposed testing scheme not only allowed for meaningful statistical benchmarking of SVM versus competitors regarding accuracy, test time, and train time; it also enabled us to quantify as to what degree our conclusions remained stable under contamination of the benchmark suite.

**Method comparison:** The results highlight the advantages of the GSD-front over existing approaches. Applying *first-order stochastic dominance* (a special case of GSD where $R_2^*$ is the trivial preorder) on the same set-up, yields that no classifier is significantly larger than (or incomparable to) any other classifier, based on a 5% significance level. This illustrates that the GSD-approach accounts for accuracy being a *cardinal* measure. In contrast, the *Pareto-front* here contains all considered classifiers. Thus, the Pareto front is much less informative than the GSD-front, which is also reflected in Theorem 2. Unlike the Pareto-front, the GSD-front is based on the distribution of the multidimensional quality metric and not only on the pairwise comparisons, and can use this knowledge to define the front. Thus, the GSD front is a balance between the conservative Pareto analysis and the liberal weighted sum comparison. Finally, we want to compare our method with an approach based on extending the test for single quality metrics proposed in [24] to the multiple metric setting. We therefore perform all possible single-metric tests as in [24] and define the *marginal front* as those classifiers that are not statistically significantly worse than another classifier on *all* metrics. However, this procedure can not be used to define a hypothesis test. Therefore, only a comparison with the empirical GSD-front is meaningful. For OpenML, this marginal front consists of all classifiers and is less exploratory than the empirical GSD-front. More details on the results of these other approaches and how these compare to the GSD front can be found in Appendix C.1.

## 5.2 Experiments on PMLB

We select 62 datasets from the Penn Machine Learning Benchmark (PMLB) suite [64] according to criteria explained in Appendix C.2. The following analysis shall exemplify how our proposed statistical tests can aid researchers in benchmarking newly developed classifiers against state-of-the-art ones. To this end, we compare a recently proposed classifier based on compressed rule ensembles of trees (CRE) [62] w.r.t. robust accuracy against five well-established classifiers, namely CART, RF, SVM with radial kernel, kNN and GLMNet. We operationalize the latent quality criterion of robust accuracy through i) classical accuracy (metric), ii) robustness of accuracy w.r.t. noisy features (ordinal), and iii) robustness of accuracy w.r.t. noisy classes (ordinal). Computation of i) is straightforward; in order to retrieve ii) and iii), we follow [92, 93] by randomly perturbing a share (here: 20 %) of both classes and features and computing the accuracy subsequently, as detailed in Appendix C.2. Since there exist competing definitions of robustness [43, 10, 72] and due to the share's arbitrary size, we treat ii) and iii) as ordinal and discretize the perturbated accuracy in the same way as for the runtimes in the openML experiments. Detailed results and visualization thereof can be found in Appendix C.2. In a nutshell, we find no evidence to reject the null of both the static and the dynamic GSD-test at significance level $\alpha = 0.05$. In particular, we do not reject any of the pairwise auxiliary tests for hypothesis pairs $(H_0^{C'}, \neg H_0^{C'})$ with $C := $ CRE and $C' \in \mathcal{C} \setminus \{$CRE$\}$) for neither $\alpha$ nor $\frac{\alpha}{5}$. Our analysis hence concludes that we cannot rule out at significance level $\alpha = 0.05$ that the newly proposed classifier CRE is dominated by the five state-of-the-art classifiers w.r.t. all compatible utility representation of the latent criterion robust accuracy.

## 5.3 Additional recommendations for the end-user

We end the section with a few brief general notes for end-users of our benchmark methodology. This should make it easy to decide whether a GSD-based analysis is appropriate in a given use-case.

1. GSD-based studies do not primarily aim to identify the best algorithm for a given benchmark suite. Often, the GSD front contains more than one element. They are rather intended for checking whether a newly proposed classifier for a certain problem class can potentially improve on the state-of-the-art classifiers, or whether it disqualifies itself from the outset.

2. GSD-based studies allow statements with inferential guarantees by providing appropriate statistical tests: Assuming an *i.i.d.* benchmark suite, a judgment about an algorithm represents a statement about an underlying population and not just this specific suite.

3. GSD-based studies enable the robustness of the results to be quantified under the deviation from the *i.i.d.* assumption: It can be checked which share of the benchmark suite may be contaminated without affecting the obtained inferential statements.

4. GSD-based studies allow algorithms to be compared w.r.t. multiple metrics simultaneously. They enable the full exploitation of the information contained in differently scaled metrics.

## 6 Concluding remarks

**Summary:** We introduced the GSD-front for multicriteria comparisons of classifiers, gave conditions for its consistent estimability and proposed a statistical test for checking if a classifier belongs to it. We illustrated our concepts using two well-established benchmark suites. The results came with threefold reliability: They included several quality metrics, representation of statistical uncertainty, and a quantification of robustness under deviations from the assumptions.

**Limitations and future research:** Two specific limitations open promising avenues: 1.) *Comparing other types of algorithms:* We restricted ourselves to comparing classifiers. However, any situation in which objects are to be compared on the basis of different (potentially differently scaled) metrics over a random selection of instances can be analyzed using these ideas. For instance, applications of our framework to the multicriteria deep learning benchmark suite DAWNBench [21] or the bi-criteria optimization benchmark suite DeepOBS [74] appear straighforward. 2.) *Extension to regression-type analysis:* Analyses based on the GSD-front do not account for meta properties of the data sets. A straightforward extension to the case of additional covariates for the data sets is to stratify by these for the GSD-comparison. This would allow for a situation-specific GSD-analysis, presumably yielding more informative results.

## Acknowledgements

We thank the anonymous reviewers and the area chair for providing valuable feedback. HB sincerely thanks the Evangelisches Studienwerk Villigst e.V. for the funding and support of her doctoral studies. Support by the Federal Statistical Office of Germany within the co-operation project "Machine Learning in Official Statistics" (JR and TA), by the Bavarian Academy of Sciences (BAS) through the Bavarian Institute for Digital Transformation (bidt, JR) and by the LMU Mentoring Program (JR and HB) is gratefully acknowledged.

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

# A  Mathematical background

## A.1  Basic definitions from order theory

A binary relation $R$ on a set $M$ is a subset of the Cartesian product of $M$ with itself, i.e., $R \subseteq M \times M$. $R$ is called *reflexive*, if $(a,a) \in R$, *transitive*, if $(a,b),(b,c) \in R \Rightarrow (a,c) \in R$, *antisymmetric*, if $(a,b),(b,a) \in R \Rightarrow a = b$, and *complete*, if $(a,b) \in R$ or $(b,a) \in R$ (or both) for arbitrary elements $a,b,c \in M$. A *preference relation* is a binary relation that is complete and transitive; a *preorder* is a binary relation that is reflexive and transitive; a *linear order* is a preference relation that is antisymmetric; a *partial order* is a preorder that is antisymmetric. If $R$ is a preorder, we denote by $P_R \subseteq M \times M$ its *strict part* and by $I_R \subseteq M \times M$ its *indifference part*, defined by $(a,b) \in P_R \Leftrightarrow (a,b) \in R \wedge (b,a) \notin R$, and $(a,b) \in I_R \Leftrightarrow (a,b) \in R \wedge (b,a) \in R$.

## A.2  Detailed description of the permutation test from Section 4.1

In this section we describe in detail the statistical test for the hypothesis pair $(H_0^{C'}, \neg H_0^{C'})$ discussed in Section 4.1 and first introduced in [48]. Moreover, we give further details on our proposed extension of this test to the global hypothesis pair $(H_0, \neg H_0)$ in both the static and the dynamic variant.

### A.2.1  Preliminaries

Before we can describe the test from Section 4.1 in detail, we first need to recall two more definitions.

**Definition 7.** *Let $\mathcal{A} = [A, R_1, R_2]$ be a preference system. We call $\mathcal{A}$ **bounded**, if there exist $a_*, a^* \in A$ such that $(a^*, a) \in R_1$, and $(a, a_*) \in R_1$ for all $a \in A$, and $(a^*, a_*) \in P_{R_1}$.*

**Definition 8.** *Let $\mathcal{A} = [A, R_1, R_2]$ be a consistent and bounded preference system with $a_*, a^*$ as before. Define*

$$\mathcal{N}_\mathcal{A} := \big\{ u \in \mathcal{U}_\mathcal{A} : u(a_*) = 0 \ \wedge \ u(a^*) = 1 \big\}.$$

*For $\delta \in [0,1)$, denote by $\mathcal{N}_\mathcal{A}^\delta$ the set of all $u \in \mathcal{N}_\mathcal{A}$ with*

$$u(a) - u(b) \geq \delta \quad \wedge \quad u(c) - u(d) - u(e) + u(f) \geq \delta$$

*for all $(a,b) \in P_{R_1}$ and for all $((c,d),(e,f)) \in P_{R_2}$.*

We now start by describing an adapted version of the permutation test for the hypothesis pairs $(H_0^{C'}, \neg H_0^{C'})$ proposed in [48]. For a concrete realization of the *i.i.d.*-sample of data sets $D_1 := T_1(\omega_0), \ldots, D_s := T_s(\omega_0) \in \mathcal{D}$ with $s \in \mathbb{N}$ associated with $\omega_0 \in \Omega$, we define the set

$$(C, C')_{\omega_0} = \{ \Phi(C, D_i) : i \leq s \} \cup \{ \Phi(C', D_i) : i \leq s \} \cup \{ \mathbf{0}, \mathbf{1} \},$$

where $\mathbf{0}$ is the vector containing $n$ zeros and $\mathbf{1}$ is the vector containing $n$ ones. Denote by $\mathcal{P}_{\omega_0}$ the restriction of $\mathcal{P}$ to $(C, C')_{\omega_0}$. It is then easy to verify that $\mathcal{P}_{\omega_0}$ is a consistent and bounded preference system with $a_* := \mathbf{0}$ and $a^* := \mathbf{1}$. For testing the hypothesis pair $(H_0^{C'}, \neg H_0^{C'})$ defined and discussed in Section 4.1 of the main text, we then use the following regularized test statistic for the specific sample induced by $\omega_0$:

$$d_s^\delta(C', C)(\omega_0) := \inf_{u \in \mathcal{N}_{\mathcal{P}_{\omega_0}}^{\mu_\delta}} \sum_{z \in (C,C')_{\omega_0}} u(z) \cdot (\hat{\pi}_{C'}^{\omega_0}(\{z\}) - \hat{\pi}_C^{\omega_0}(\{z\}))$$

with $\delta \in [0,1]$ and $\mu_\delta := \delta \cdot \sup\{ \xi : \mathcal{N}_{\mathcal{A}_\omega}^\xi \neq \emptyset \}$, and $\hat{\pi}_C^{\omega_0}$ resp. $\hat{\pi}_{C'}^{\omega_0}$ are the empirical probability measures of the performances of $C$ resp. $C'$ for the specific sample induced by $\omega_0$.

### A.2.2  Testing scheme for $(H_0^{C'}, \neg H_0^{C'})$

We denote our samples as follows:

$$
\begin{aligned}
\mathbf{x} &:= (x_1, \ldots, x_s) := (\Phi(C, D_1), \ldots, \Phi(C, D_s)) \\
\mathbf{y} &:= (y_1, \ldots, y_s) := (\Phi(C', D_1), \ldots, \Phi(C', D_s))
\end{aligned}
$$

The concrete testing scheme for the permutation test for hypothesis pair $(H_0^{C'}, \neg H_0^{C'})$ then looks as follows:

**Step 1:** Take the pooled data sample: $\mathbf{w} := (w_1, \ldots, w_{2s}) := (x_1, \ldots, x_s, y_1, \ldots, y_s)$

**Step 2:** Take all $r := \binom{2s}{s}$ index sets $I \subseteq \{1, \ldots, 2s\}$ of size $s$. Evaluate $d_s^\delta(C', C)$ for $(w_i)_{i \in I}$ and $(w_i)_{i \in \{1, \ldots, n+m\} \setminus I}$ instead of $\mathbf{x}$ and $\mathbf{y}$. Denote the evaluations by $d_I^\delta$.

**Step 3:** Sort all $d_I^\delta$ in increasing order to get $d_{(1)}^\delta, \ldots, d_{(r)}^\delta$.

**Step 4:** Reject $H_0^{C'}$ if $d_s^\delta(C', C)(\omega_0)$ is strictly smaller than $d_{(\ell)}^\delta$, with $\ell := \lfloor \alpha \cdot r \rfloor$ and $\alpha$ the significance level.

Note that, for large $\binom{2s}{s}$, we can approximate the above resampling scheme by computing $d_I^\delta$ only for a large number $N$ of randomly drawn $I$. Moreover, note that only the *i.i.d.* assumption is needed for the above test to be valid.

### A.2.3 Static GSD-test

As argued in the Section 4.1 of the main part of the paper, if we want to obtain a valid statistical test at the significance level $\alpha \in [0, 1]$ for hypothesis pair $(H_0, \neg H_0)$, we can simply perform all pairwise tests of hypothesis pairs $(H_0^{C'}, \neg H_0^{C'})$ at this same significance level $\alpha$. We can then reject the hypothesis $H_0$ at level $\alpha$ if we can reject each hypothesis $H_0^{C'}$ at level $\alpha$ or, in other words, if

$$\min \left\{ \frac{1}{N} \cdot \sum_{I \in \mathcal{I}_N} \mathbb{1}_{\left\{ d_s^\delta(C', C)(\omega_0) < d_I^\delta \right\}} : C' \in \mathcal{C} \setminus \{C\} \right\} \geq 1 - \alpha.$$

We call this the static GSD-test.

To see that this procedure indeed gives a valid level-$\alpha$ test for the global hypothesis pair $(H_0, \neg H_0)$, observe that – assuming $H_0$ to be true – the probability of $H_0$ being rejected equals the probability of *all* hypothesis $H_0^{C'}$ being rejected simultaneously. The latter probability – still assuming $H_0$ to be true – is obviously bounded from above by the probability that *one specific* hypothesis $H_0^{C^*}$ is rejected, which itself is bounded from above by the significance level $\alpha$ by construction.

### A.2.4 Dynamic GSD-test

As discussed in the main text and reprinted here again for convenience of the reader, a slightly modified test in the context of the GSD-front is directly derivable: If one is rather interested in identifying the maximal subset $\mathcal{S}_{\max}$ of $\mathcal{C}$ for which $C$ significantly lies in the GSD-front, i.e., in testing the hypothesis pairs $(\tilde{H}_0^\mathcal{S}, \neg \tilde{H}_0^\mathcal{S})$ for all $\mathcal{S} \subseteq \mathcal{C}$ *simultaneously*, the following alternative test would be a statistically valid level-$\alpha$ test: First, perform all individual tests for $(H_0^{C'}, \neg H_0^{C'})$ with level $\frac{\alpha}{c}$. Then identify $\mathcal{S}_{\max}$ as the set of all classifiers from $\mathcal{C}$ for which the individual hypotheses are rejected. The (random) alternative hypothesis $\tilde{H}_1^{\mathcal{S}_{\max}} : C \in \mathrm{gsd}(\mathcal{S}_{\max})$ is then statistically valid in the sense of being false only with a probability bounded by $\alpha$. We call this the dynamic GSD-test.

To see that this procedure indeed gives a valid level-$\alpha$ test for the (random) hypothesis pair $(\tilde{H}_0^{\mathcal{S}_{\max}}, \neg \tilde{H}_0^{\mathcal{S}_{\max}})$, observe that – under the null hypothesis – the probability of $C$ lying in the GSD-front of some random subset $\mathcal{S}$ of $\mathcal{C}$ is bounded from above by the sum of probabilities of $C$ lying in the GSD-front of $\{C, S\}$, where summation is over all $S \in \mathcal{S}$. As each of these probabilities is bounded from above by $\frac{\alpha}{c}$ by construction, the corresponding sum is bounded from above by $|\mathcal{S}| \cdot \frac{\alpha}{c}$. Finally, as $|\mathcal{S}| \leq c$, this gives the desired upper bound of $\alpha$.

### A.2.5 Computation and regularization

Note that the test statistic $d_s^\delta(C', C)(\omega_0)$ can be computed by solving a linear optimization problem (see [48, Proposition 4]) and, hence, the test just described is computationally tractable.

Moreover, note that in both applications in Section 5 the tests are based on the unregularized statistics $d_s^0(C', C)$, as the regularization performed in [48] aims at reaching a goal which is not primarily relevant for the present paper: The authors there are primarily interested in significantly detecting GSD of one variable over the other. Consequently, their regularization aims at making the test more sensitive for exactly this purpose. In contrast, in our study we are primarily interested in significantly detecting *incomparabilities* between variables, making the regularization by far less natural.

## A.3 Detailed description of the robustness check from Section 4.2

In this section we describe in detail the robustification of the statistical test for the hypothesis pair $(H_0^{C'}, \neg H_0^{C'})$ discussed in Section 4.2 and first introduced in [48]. Moreover, we give further details on our proposed extension of this robustified statistical test to the global hypothesis pair $(H_0, \neg H_0)$ in both the static and the dynamic variant.

### A.3.1 Preliminiaries

If we assume, as done in Section 4.2, that up to $k \leq s$ of the observations in our sample $T_1, \ldots, T_s$ might be contaminated and, accordingly, follow any arbitrary distribution, then we have to base the permutation test for hypothesis pair $(H_0^{C'}, \neg H_0^{C'})$ on a worst-case analysis of between the measures contained in $\mathcal{M}_C$ and $\mathcal{M}_{C'}$ defined instead of the true empirical measures of the two samples induced by the classifiers $C$ and $C'$. If again $D_1 := T_1(\omega_0), \ldots, D_s := T_s(\omega_0) \in \mathcal{D}$ is a concrete (now potentially contaminated) sample associated with $\omega_0 \in \Omega$, and we again define $\mathbf{x}$ and $\mathbf{y}$ as in Section A.2, then the observed contamination models of $C$ and $C'$ look as follows:

$$\mathcal{M}_C(\omega_0) = \left\{ (1 - \tfrac{k}{s})\hat{\pi}_C^{cont,\omega_0} + \tfrac{k}{s}\mu : \mu \text{ probability measure} \right\},$$

$$\mathcal{M}_{C'}(\omega_0) = \left\{ (1 - \tfrac{k}{s})\hat{\pi}_{C'}^{cont,\omega_0} + \tfrac{k}{s}\mu : \mu \text{ probability measure} \right\}.$$

### A.3.2 Testing scheme for robustified test on $(H_0^{C'}, \neg H_0^{C'})$

If we set

$$\overline{d_s^\delta}(C', C)(\omega_0) := \sup_{\pi_1 \in \mathcal{M}_{C'}(\omega_0), \pi_2 \in \mathcal{M}_C(\omega_0)} \left( \inf_{u \in \mathcal{N}_{\mathcal{P}\omega_0}'^{\mu\delta}} \sum_{z \in (C,C')_{\omega_0}} u(z) \cdot (\pi_1(\{z\}) - \pi_2(\{z\})) \right),$$

$$\underline{d_s^\delta}(C', C)(\omega_0) := \inf_{\pi_1 \in \mathcal{M}_{C'}(\omega_0), \pi_2 \in \mathcal{M}_C(\omega_0)} \left( \inf_{u \in \mathcal{N}_{\mathcal{P}\omega_0}'^{\mu\delta}} \sum_{z \in (C,C')_{\omega_0}} u(z) \cdot (\pi_1(\{z\}) - \pi_2(\{z\})) \right),$$

then the concrete testing scheme for the permutation test for hypothesis pair $(H_0^{C'}, \neg H_0^{C'})$ *under at most $k$ contaminated sample members* looks as follows:

**Step 1:** Take the pooled data sample: $\mathbf{w} := (w_1, \ldots, w_{2s}) := (x_1, \ldots, x_s, y_1, \ldots, y_s)$

**Step 2:** Take all $r := \binom{2s}{s}$ index sets $I \subseteq \{1, \ldots, 2s\}$ of size $s$. Evaluate $\underline{d_s^\delta}(C', C)$ for $(w_i)_{i \in I}$ and $(w_i)_{i \in \{1, \ldots, n+m\} \setminus I}$ instead of $\mathbf{x}$ and $\mathbf{y}$. Denote the evaluations by $\underline{d_I^\delta}$.

**Step 3:** Sort all $\underline{d_I^\delta}$ in increasing order to get $\underline{d^\delta}_{(1)}, \ldots, \underline{d^\delta}_{(r)}$.

**Step 4:** Reject $H_0^{C'}$ if $\overline{d_s^\delta}(C', C)(\omega_0)$ is strictly smaller than $\underline{d^\delta}_{(\ell)}$, with $\ell := \lfloor \alpha \cdot r \rfloor$ and $\alpha$ the significance level.

The adapted testing scheme just described gives a valid (yet conservative) level-$\alpha$-test for the hypothesis pair $(H_0^{C'}, \neg H_0^{C'})$ under at most $k$ contaminated sample members.

Moreover, it directly follows from the discussions in Part C of the supplementary material to [48] that the (approximate) observed p-value of this test is given by

$$f_{(C',C)}(k) := 1 - \tfrac{1}{N} \cdot \sum_{I \in \mathcal{I}_N} \mathbb{1}_{\left\{ d_I^\delta - d_s^\delta(C',C)(\omega_0) > \frac{2k}{(s-k)} \right\}},$$

where again $N$ denotes the number of resamples, $\mathcal{I}_N$ is the corresponding set of resamples, and $d_I^\delta$ is the test statistic evaluated for the resample associated to $I$.

### A.3.3 Robustified static GSD-test

As already argued in the main text, it is now easy to calculate an (approximate) observed p-value for our global hypothesis pair $(H_0, \neg H_0)$: We simply calculate the maximal observed p-value among all $C' \in \mathcal{C} \setminus \{C\}$, i.e. set

$$F_C(k) := \max\{ f_{(C',C)}(k) : C' \in \mathcal{C} \setminus \{C\} \}.$$

The robustified test for the degree of contamination $k$ can be carried out as follows: Calculate $F_C(k)$ and reject $H_0$ if $F_C(k) \leq \alpha$, i.e., if the maximal (approximate) $p$-value of the pairwise tests is still lower or equal than the significance level.

The argument that the testing procedure just described indeed produces a valid level-$\alpha$ test of the global hypothesis pair $(H_0, \neg H_0)$ under up to $k$ contaminated data sets in the sample, can be carried out completely analogous as done in Appendix A.2.3.

### A.3.4  Robustified dynamic GSD-test

Finally, as discussed in the main text and reprinted here again for convenience of the reader, also the robustified dynamic GSD-test can be obtained in a straightforward manner: Under up to $k$ contaminated data sets in the sample, the (random) alternative hypothesis $\tilde{H}_1^{\mathcal{S}_{\max}} : C \in \mathrm{gsd}(\mathcal{S}_{\max})$ from before is statistically valid with level $\alpha$ if all individual robustified tests reject $H_0^{C'}$ at level $\frac{\alpha}{c}$, i.e., if $F_C(k) \leq \frac{\alpha}{c}$.

The argument that the testing procedure just described indeed produces a valid level-$\alpha$ test for the (random) hypothesis pair $(\tilde{H}_0^{\mathcal{S}_{\max}}, \neg \tilde{H}_0^{\mathcal{S}_{\max}})$ under up to $k$ contaminated data sets in the sample, can be carried out completely analogous as done in Appendix A.2.4.

### A.3.5  Computation and regularization

Note that also the robustified test statistic $\overline{d_s^\delta}(C', C)(\omega_0)$ can be computed by solving a linear optimization problem (see [48, Proposition 6]) and, hence, the test just described is computationally tractable.

Again, note that the tests in Section 5 are based on the unregularized test statistics with $\delta = 0$. The reason for this is the same as discussed at the end of Appendix A.2.

## B  Proofs

### B.1  Proof of Theorem 1

**Proof.** First, note that for $C, C' \in \mathcal{C}$, we have that $C \succsim C'$ if and only if

$$D(C, C') := \inf_{u \in \mathcal{U}_{\mathcal{P}_\Phi}} \left( \mathbb{E}_\pi(u \circ \Phi_C) - \mathbb{E}_\pi(u \circ \Phi_{C'}) \right) \geq 0.$$

Thus, the GSD-front can equivalently be rewritten as

$$\mathrm{gsd}(\mathcal{C}) = \left\{ C \in \mathcal{C} : \nexists C' \in \mathcal{C} \text{ s.t. } \begin{matrix} D(C', C) \geq 0 \\ D(C, C') < 0 \end{matrix} \right\}.$$

Now, let $\varepsilon : \mathbb{N} \to [0, 1] : s \mapsto 1/\sqrt[4]{s}$. We show that:

$$C \in \mathrm{gsd}(\mathcal{C}) \Rightarrow C \in \lim_{s \to \infty} \mathrm{egsd}_s^{\varepsilon(s)}(\mathcal{C}) \quad \pi\text{-a.s.} \text{, and} \tag{3}$$

$$C \notin \mathrm{gsd}(\mathcal{C}) \Rightarrow C \notin \lim_{s \to \infty} \mathrm{egsd}_s^{\varepsilon(s)}(\mathcal{C}) \quad \pi\text{-a.s.} \tag{4}$$

Note that the proof immediately translates to the more general case of $\varepsilon(s) \in \Theta(1/\sqrt[4]{s})$ as stated in Theorem 1. Denote with $\hat{\mathbb{E}}$ the expectation w.r.t. the empirical measure associated with the i.i.d. sample[10] $(T_1, \ldots, T_s)$. For Implication (3), assume that $C \in \mathrm{gsd}(\mathcal{C})$. Then for every other classifier $C'$ there exists an utility function $u \in \mathcal{U}_{\mathcal{P}_\Phi}$ with $\mathbb{E}_\pi(u \circ \Phi_C) > \mathbb{E}_\pi(u \circ \Phi_{C'})$ (Otherwise we would have $D(C', C) \geq 0$ and $D(C, C') < 0$, where the second statement is due to antisymmetry). For these corresponding utility functions, because of the strong law of large numbers, we would get $d_s(C', C) \leq \hat{\mathbb{E}}(u \circ \Phi_{C'}) - \hat{\mathbb{E}}(u \circ \Phi_C) \xrightarrow{a.s.} c < 0$. Since $\mathcal{C}$ consists only of finitely many classifiers, $\mathrm{egsd}_s^{\varepsilon(s)}(\mathcal{C})$ will almost surely not contain $C$ asymptotically. Note that for Implication (3) to hold,

---

[10]Note that assuming only an exchangeable sample would also suffice. Note further that we have to assume the measurability of the involved infimum type statistics. For more details on this issue, see, e.g., [28].

it is only necessary that $\varepsilon(s)$ converges to zero as $s$ goes to infinity. The order of convergency as $\Theta(1/\sqrt[4]{s})$ is only needed for Implication (4).

For Implication (4) assume that $C \notin \mathrm{gsd}(\mathcal{C})$. Then there exists a classifier $C'$ with $D(C', C) \geq 0$ and $D(C, C') < 0$. An analog argumentation like above shows that $d_s(C, C')$ converges almost surely to a value smaller than zero. It remains to analyze $D(C', C)$. For this, we have to show that $d_s(C', C) + \varepsilon(s) \xrightarrow{a.s.} c \geq 0$. We utilize uniform convergence: For arbitrary $\xi > 0$, [84, p. 192 Theorem 5.1] gives us

$$P\left(\sup_{u \in \mathcal{U}_{\mathcal{P}_\Phi}} \left|\mathbb{E}(u \circ \Phi_C) - \hat{\mathbb{E}}(u \circ \Phi_C)\right| > \xi\right) \leq 8\left(\frac{e \cdot 2s}{h}\right)^h \cdot \exp\left\{-\xi_*^2 s\right\},$$

where $\xi_* = \xi - 1/s$ and $h$ is the VC dimension of $\mathcal{I}_\Phi$. The same holds for $\Phi_{C'}$. The triangle inequality then gives

$$P\left(\sup_{u \in \mathcal{U}_{\mathcal{P}_\Phi}} \left|\hat{\mathbb{E}}(u \circ \Phi_C) - \hat{\mathbb{E}}(u \circ \Phi_{C'})\right| > 2\xi\right) \leq 8\left(\frac{e \cdot 2s}{h}\right)^h \cdot \exp\left\{-\xi_*^2 s\right\}.$$

For $\varepsilon(s) = 1/\sqrt[4]{(1/s)}$ and $s$ large enough, we have $\varepsilon_*(s) = \varepsilon(s) - 1/s \geq \varepsilon(s)/2$ and therefore

$$P\left(\sup_{u \in \mathcal{U}_{\mathcal{P}_\Phi}} \left|\hat{\mathbb{E}}(u \circ \Phi_C) - \hat{\mathbb{E}}(u \circ \Phi_{C'})\right| > 2\varepsilon(s)\right) \leq 8\left(\frac{e \cdot 2s}{h}\right)^h \cdot \exp\left\{-\varepsilon_*(s)^2 s\right\}$$

$$\leq 8\left(\frac{e \cdot 2s}{h}\right)^h \exp\left\{-\varepsilon(s)^2 s/4\right\}.$$

This implies

$$P\left(\sup_{u \in \mathcal{U}_{\mathcal{P}_\Phi}} \left|\hat{\mathbb{E}}(u \circ \Phi_C) - \hat{\mathbb{E}}(u \circ \Phi_{C'})\right| > \varepsilon(s)/2\right) \leq 8\left(\frac{e \cdot 2s}{h}\right)^h \exp\left\{-\varepsilon(s)^2 s/64\right\} \quad (5)$$

$$= 8\left(\frac{e \cdot 2s}{h}\right)^h \exp\left\{-\sqrt{s}/64\right\}. \quad (6)$$

If the VC dimension $h$ is finite, the term $8\left(\frac{e \cdot 2s}{h}\right)^h$ is polynomially growing in $\sqrt{s}$ (or $s$), whereas the term $\exp\left\{-\sqrt{s}/64\right\}$ is exponentially decreasing in $\sqrt{s}$ (or $s$). Therefore, the right hand side of Inequality (5) converges to zero, which shows that

$$\sup_{u \in \mathcal{U}_{\mathcal{P}_\Phi}} \left|\hat{\mathbb{E}}(u \circ \Phi_C) - \hat{\mathbb{E}}(u \circ \Phi_{C'})\right| - \varepsilon(s)$$

converges in probability to a value $c \leq 0$ or equivalently, that $d_s(C', C) + \varepsilon(s)$ converges to a value $c \geq 0$. Since the right hand side of Inequality (5) converges exponentially in $s$, the Borel-Cantelli theorem (cf., e.g., [29, p.67ff]) gives also strong convergency, which completes the proof. Note that it is not necessary to specify $\varepsilon(s)$ concretely as $1/\sqrt[4]{s}$. It would be sufficient to define $\varepsilon(s)$ as of the order of $\Theta(1/\sqrt[4]{s})$. $\qquad\square$

## B.2 Proof of Theorem 2

**Proof.** i) Assume that $C \notin \mathrm{par}(\Phi)$. Then, by definition of $\mathrm{par}(\Phi)$, there exists $C' \in \mathcal{C}$ such that for all $D \in \mathcal{D}$ it holds that $\Phi(C', D) \succ \Phi(C, D)$. This implies that for all $D \in \mathcal{D}$ it holds that $(\Phi(C', D), \Phi(C, D)) \in P_{R_1^*}$. Now, choose $u \in \mathcal{U}_{\mathcal{P}_\Phi}$. Since $u$ then, by definition, is strictly isotone with respect to $P_{R_1^*}$, this allows us to conclude that the function $u(\Phi(C', \cdot)) - u(\Phi(C, \cdot))$ is strictly positive, i.e., we have $u(\Phi(C', D)) - u(\Phi(C, D)) > 0$ for arbitrary $D \in \mathcal{D}$.

We compute:

$$
\begin{aligned}
\mathbb{E}_\pi(u \circ \Phi_{C'}) - \mathbb{E}_\pi(u \circ \Phi_C) &= \int_\Omega u(\Phi(C', T(\omega)))d\pi(\omega) - \int_\Omega u(\Phi(C, T(\omega)))d\pi(\omega) \\
&= \int_\Omega \underbrace{u(\Phi(C', T(\omega))) - u(\Phi(C, T(\omega)))}_{>0 \text{ for all } \omega \in \Omega, \text{ since } T(\omega) \in \mathcal{D}} d\pi(\omega) > 0
\end{aligned}
$$

This gives $\mathbb{E}_\pi(u \circ \Phi_{C'}) > \mathbb{E}_\pi(u \circ \Phi_C)$. As $u$ was chosen arbitrarily, this implies that $C' \succ C$. Hence, by definition of the GSD-front, we have $C \notin \text{gsd}(\mathcal{C})$.

ii) First, note that both postulates are statements involving random sets (i.e., sets dependent on the realizations of the variables $T_1, \ldots, T_s$). Thus, we have to prove both statements for arbitrary realizations of these variables. So let $D_1 := T_1(\omega_0), \ldots, D_s := T_s(\omega_0) \in \mathcal{D}$ be an arbitrary realisation. For this concrete realization of the sample, the first statement is immediate, since if there is no $C'$ such that $d_s(C', C)(\omega_0) \geq -\varepsilon_2$ there is also no $C'$ such that $d_s(C', C)(\omega_0) \geq -\varepsilon_1$ (as the latter is harder to satisfy due to $\varepsilon_1 \leq \varepsilon_2$).

Again for the chosen concrete realization of the sample, the second postulate is an immediate consequence of statement i) from above. As in both situations the realization of the variables was chosen arbitrarily, this implies the statement. $\quad\square$

## B.3 Proof of Theorem 3

To see that the static test is a valid level-$\alpha$ test for the global hypothesis pair $(H_0, \neg H_0)$, observe that – assuming $H_0$ to be true – the probability of $H_0$ being rejected equals the probability of *all* hypothesis $H_0^{C'}$ being rejected simultaneously. The latter probability – still assuming $H_0$ to be true – is obviously bounded from above by the probability that *one specific* hypothesis $H_0^{C^*}$ is rejected, which itself is bounded from above by the significance level $\alpha$ by construction.

Furthermore, the reason for the consistency of the static test is the following: First, note that under the assumption of Theorem 1 (because of the finite VC dimension), we have that $d_s(C', C)$ converges to $D(C', C)$ in probability for every abrbitrary classifier $C' \neq C$. Therefore, for fixed $C'$ and under the null hypothesis $H_0^{C'}$, we have $d_s(C', C)$ converges in probability to a value larger than or equal to zero. This implies that under this null hypothesis the implicit critical values of the permutation test become arbirarily close to a values larger than or equal to zero.

Now, let $C$ be in the GSD-front. Then, due to antisymmetry of $\succsim$, for every other classifier $C'$, there exists a utility for which the expectation of $u \circ \Phi_C$ is larger than the expectation of $u \circ \Phi_{C'}$. Because of the weak law of large numbers, this translates to the empirical expectations with an arbitrarily high probability if only the sample size is large enough. Therefore, all-together, as $s$ converges to infinity, the test rejects the null hypothesis in this situation with arbitrary high probability. Finally, since we have only a finite number of hypothesis of the static test, this also translates to the static test itself. Therefore the static test is indeed a consistent level-$\alpha$ test.

To see that also the dynamic test is a valid level-$\alpha$ test for the (random) hypothesis pair $(\tilde{H}_0^{\mathcal{S}_{\max}}, \neg\tilde{H}_0^{\mathcal{S}_{\max}})$, observe that – under the null hypothesis – the probability of $C$ lying in the GSD-front of some random subset $\mathcal{S}$ of $\mathcal{C}$ is bounded from above by the sum of probabilities of $C$ lying in the GSD-front of $\{C, S\}$, where summation is over all $S \in \mathcal{S}$. As each of these probabilities is bounded from above by $\frac{\alpha}{c}$ by construction, the corresponding sum is bounded from above by $|\mathcal{S}| \cdot \frac{\alpha}{c}$. Finally, as $|\mathcal{S}| \leq c$, this gives the desired upper bound of $\alpha$.

Finally, also the consistency of the dynamic test follows from the fact that it is constructed from a finite set of consistent tests for every possible set $\mathcal{S} \subseteq \mathcal{C}$. $\quad\square$

## B.4 Proof of Corollary 1

Assume that $\Phi(\mathcal{C} \times \mathcal{D}) \subseteq M \times S_1 \times S_2$, where $S_1, S_2 \subset [0, 1]$ are finite, and $M \subseteq [0, 1]$ is arbitrary. This is possible since by definition of $\Phi$ we have $M \subseteq \phi_1(\mathcal{C} \times \mathcal{D})$ and $S_1 \subseteq \phi_2(\mathcal{C} \times \mathcal{D})$ and $S_2 \subseteq \phi_3(\mathcal{C} \times \mathcal{D})$, and the metrics $\phi_2$ and $\phi_3$ are assumed to be finitely discrete. We show that the width[11] of the restriction of $R_1^*$ to $\Phi(\mathcal{C} \times \mathcal{D})$ is finite. It then follows directly from e.g. [76, Proposition 2] that the VC-dimension of

$$\mathcal{I}_\Phi := \left\{ \{a : u(a) \geq c\} : c \in [0, 1] \wedge u \in \mathcal{U}_{\mathcal{P}_\Phi} \right\}$$

is also finite. The claim then follows from Theorem 1.

---

[11]The *width* of a preordered set is the maximal cardinality of an antichain, i.e., the maximal number of pairwise incomparable elements.

To show the finiteness of the width, assume - wlog - that $|S_1| = g < \infty$ and $|S_2| = h < \infty$. Assume, for contradiction, that there exists an antichain[12] $Q \subseteq \Phi(\mathcal{C} \times \mathcal{D})$ within the restriction of $R_1^*$ to $\Phi(\mathcal{C} \times \mathcal{D})$ of cardinality strictly greater than $g \cdot h$. Then there exist $x = (x_1, x_2, x_3), y = (y_1, y_2, y_3) \in Q$ such that $x_2 = y_2$ and $x_3 = y_3$ (as there are only $g \cdot h$ different combinations of the second and the third component). However, since the first component is completely ordered by $\geq$, this implies either $(x, y)$ or $(y, x)$ (or both) is contained in the restriction of $R_1^*$ to $\Phi(\mathcal{C} \times \mathcal{D})$. This is a contradiction to $x$ and $y$ being elements of the same antichain $Q$, completing the argument. $\quad\square$

## C   Further results on the applications

This section provides further information on the benchmarking examples in Section 5.

### C.1   Experiments with OpenML

This sections gives further insight to the example on the OpenML data analysed in Section 5.1. We start with giving more details on the data set with all the computation settings of the classifier algorithms. Afterwards, we provide more graphics and explanations of the analysis.

#### C.1.1   Data

Overall, we are comparing the performance of *Support Vector Machine* (SVM) to further 6 classifier algorithms on 80 data sets. The data sets as well as the performance evaluation is given by the OpenML library [82].[13] The analysis is restricted to binary classification problems. We selected those data sets of OpenML that evaluated the *predictive accuracy*, *train data time computation* and *test data time computation* (both measured in milliseconds) for all of the 7 algorithms. Since the computation times depend on the environment, i.e. the number of cores used or the free memory, we discretized the computation times and considered them as ordinal. Therefore, we divided each computation time into ten categories, where category one contains the 10% highest times, and so on. Moreover, we restricted our analysis on data sets with more than 450 and less than 10000 observations. This gives us in total 80 data sets.

The algorithms discussed are:

- *Support Vector Machine* (SVM) algorithm is implemented in the `e1071` library [26]
- *Random Forests* (RF) algorithm is implemented in the `ranger` library [88],
- *Decision Tree* (CART) algorithm is implemented via the `rpart` library [79],
- *Logistic regression* (LR) algorithm is implemented via the `nnet` library [68],
- *eXtreme Gradient Boosting* (xGBoost) algorithm is implemented in the `xgboost` library [20],
- *Elastic net* (GLMNet) algorithm is implemented through the `glmnet` library [32], and
- *k-nearest neighbors* (kNN) algorithm is implemebted via the `kknn` library [38].

#### C.1.2   Detailed results of the GSD-based analysis

We started our analysis in Section 5.1 by computing the empirical GSD-front. This gives the Hasse graph 2, where a top-down edge from $C$ to $C'$ states that $d_{80}(C, C') \geq 0$ holds.

In addition to the left of Figures 1 (densities of resampled test statistics) and the right of Figure 1 (effect of contamination on p-values) in the main paper, we include the cumulative distribution functions (CDFs) in Figure 3. Since we do not include the values of the observed test statistics here, the differences in distributions are visible to a greater extent. We observe the resampled test statistics' distributions for SVM vs. xGBoost and GLMNet to be left-shifted compared to SVM vs. CART, xGBoost, and LR. A visual analysis of the test decision, however, is not possible in the absence of the observed test statistics. This is why we include their values in the caption of Figure 3.

---

[12]An *antichain* of a preordered set $(M, R)$ is a subset $A \subseteq M$ such that for all $m_1, m_2 \in A$ it holds $(m_1, m_2) \notin R$ and $(m_2, m_1) \notin R$.

[13]Last OpenML access: 24/10/2024

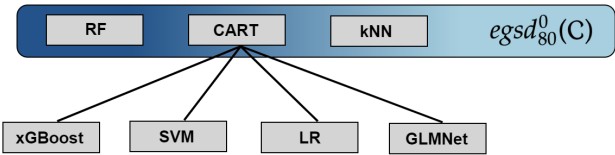

Figure 2: The blue shaded region symbolizes the $0$-empirical GSD-front for the OpenML data sets.

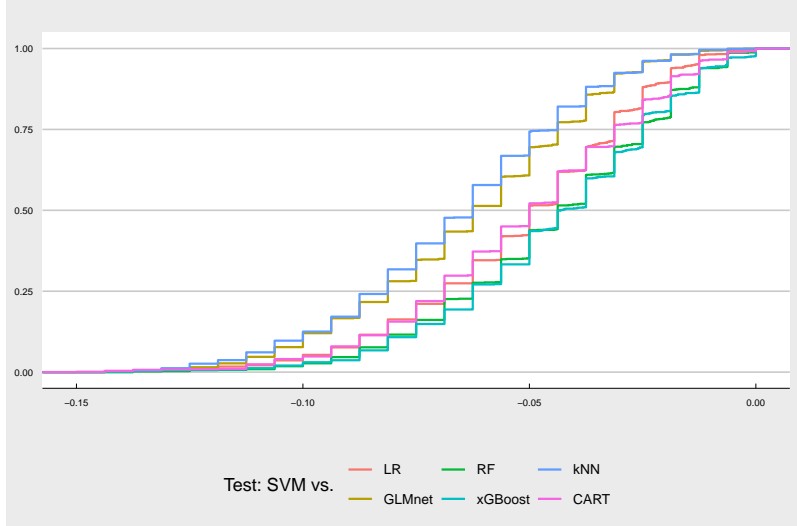

Figure 3: Cumulative Distribution Functions (CDFs) of resampled test statistics for hypothesis tests of SVM vs. LR, RF, kNN, GLMNet, xGBoost, and CART, respectively, on OpenML's benchmarking suite. As opposed to Figure 1 in the main paper, values of observed test statistics are not included. They are: $0.0125$ (CART), $-0.3875$ (kNN), $-0.4375$ (xGBoost), $-0.41875$ (RF), $-0.3375$ (GLM-Net), and $-0.04897227$ (LR). It becomes evident that the resampled test statistics' distributions for SVM vs. xGBoost and GLMNet are left-shifted compared to SVM vs. CART, xGBoost, and LR. This is also visible in Figure 1 in the main paper, albeit less clearly.

### C.1.3 Detailed results of state-of-the-art analyses and comparison to GSD-front

This section provides the detailed computation of the state-of-the-art approaches and the comparison with the GSD approach. Here, we go step by step through all the methods touched in Section C.1.

**First-order stochastic dominance** Analogously to the GSD-front, one can define the front based on (multivariate) first-order stochastic dominance (see, e.g., [77]). Note that classical first-order stochastic dominance is a special case of our generalized stochastic dominance (GSD) in the case that all quality metrics are (treated as) of ordinal scale of measurement. Given the test logic followed by, for example, [6], for the OpenML data it turns out that no classifier is significantly stochastically larger than (or incomparable to) any other classifier (based on a significance level of 5 %). Compared to the results we obtain from our GSD-front analysis, this indiscriminative result is much less informative.

**Pareto-front** Both for the PMLB benchmark suite and the OpenML setup the Pareto-front contains all considered classifiers and is therefore not very informative. This shows the advantage of our approach because our approach is generally more informative (see Theorem 2). In particular, by using generalized stochastic dominance one refrains from solely relying on pointwise comparisons of classifiers over datasets. Instead one only looks at the distribution of the multidimensional quality metric. Beyond this, compared to both a Pareto analysis and a classical first order stochastic dominance analysis (see above), the GSD approach does justice to the fact that the dimension accuracy is cardinal and, at the same time, the fact that the other dimensions are of ordinal scale of measurement. Additionally, compared to an approach that only looks at the marginal distributions of every single quality metric separately, the GSD approach takes also the dependence structure

between the different quality metrics into account. This is of particular interest if one has different performance dimensions that are anticorrelated.

**Weighted sum approach** The GSD-based approach has advantages over an approach of weighted summation of the various quality metrics especially when it is not clear how specifically the weights are to be chosen. Specifically, each weighting leads to a total ordering among the classifiers under consideration. A clear best classifier can, therefore, be identified for each weighting. Thus, different weightings generally lead to different best classifiers. As a consequence, if one chooses a specific weighting, one should really be convinced that domain knowledge thoroughly justifies it, as even small changes in the weighting can completely change the resulting ranking. In contrast, the GSD-based approach can be used if no weighting of the involved metrics is available, but still more information (e.g., from the cardinal metrics) is available than required for a Pareto-type analysis. In summary, we emphasize that our method and the weighted summation should be used under different conditions and, therefore, complement rather than compete with each other.

**Marginal-front** A highly popular testing scheme for benchmark analysis is the one proposed by [24]. We compare our approach against using a marginal front that directly results from following this scheme. This marginal front is defined as a function of (a) statistical test(s), i.e. classifiers are in it depending on the test results. We emphasize that this front is not directly comparable to the GSD-front, since the GSD-front is a theoretical object (like the Pareto front) that can be used to formulate hypotheses that can then be tested by statistical tests like the ones proposed in this paper. Thus, we compare the marginal front to the *empirical* GSD front as reported in Fig. 1 of the paper for the application on OpenML.

We run multiple single-objective evaluations and include in the marginal-front the classifiers that are not statistically significantly worse than another classifier on all metrics. For the single-objective tests, we follow the well-established procedure of [24]. That is, we first run a global Friedman test (see [33]) for the null hypothesis that all classifiers have no differences with respect to the quality metric under investigation. In case we reject this null hypothesis, we can run post hoc Nemenyi pairwise tests (see [63]), comparing the performance of algorithms pairwise, with the null hypothesis being that there is no difference between their performances w.r.t. the multidimensional quality metric considered. We would like to emphasize that such an approach does not take into account the dependence structure among the quality metrics. In other words, it only considers the marginal distribution (hence the term *marginal-front*) of the classifiers w.r.t. the individual quality metrics separately, not their joint distribution. In the following, we conduct the suggested marginal analysis for OpenML w.r.t. the three-dimensional quality metric considered (accuracy, computation time on training data, computation time on test data):

**Accuracy**

- *Global Friedman Test:* Friedman rank sum test [33] rejects global null of no differences (p-value = 3.986e-14). This means we can conduct (two-sided) pairwise post hoc tests ($\alpha = 0.05$) with no difference as the null hypothesis.

- *Post Hoc Nemenyi Test:* Table 1 below shows the pairwise comparisons of algorithm performance with the Nemenyi test [63]. P-values below 0.05 are highlighted and indicate statistically significant differences in performance.

Table 1: Pairwise comparisons of algorithm performance with the Nemenyi test based on accuracy. Underlined values indicate differences significant at $\alpha = 0.05$ level.

|         | LR         | RF         | CART    | SVM       | xGBoost | GLMNet  | kNN |
|---------|------------|------------|---------|-----------|---------|---------|-----|
| RF      | $3.9e-11$  | -          | -       | -         | -       | -       | -   |
| CART    | 0.19662    | $6.9e-05$  | -       | -         | -       | -       | -   |
| SVM     | 0.00055    | 0.06513    | 0.55264 | -         | -       | -       | -   |
| xGBoost | 0.92896    | $5.7e-08$  | 0.85263 | 0.03259   | -       | -       | -   |
| GLMNet  | 0.92341    | $6.4e-08$  | 0.86095 | 0.03446   | 1.00000 | -       | -   |
| kNN     | 0.98454    | $9.2e-09$  | 0.68760 | 0.01261   | 0.99995 | 0.99993 | -   |

**Computation time on training data**

- *Global Friedman Test:* Friedman rank sum test [33] rejects the global null hypothesis of no differences (p-value < 2.2e-16). This means we can conduct pairwise post hoc tests ($\alpha = 0.05$) with the null hypothesis of no difference.
- *Post Hoc Nemenyi Test* [63], see Table 2.

Table 2: Pairwise comparisons of algorithm performance with the Nemenyi test based on computation time on the training data. Underlined values indicate differences significant at $\alpha = 0.05$ level.

|         | LR          | RF          | CART        | SVM         | xGBoost     | GLMNet      | kNN |
|---------|-------------|-------------|-------------|-------------|-------------|-------------|-----|
| RF      | $9.1e-14$   | -           | -           | -           | -           | -           | -   |
| CART    | $0.13788$   | $< 2e-16$   | -           | -           | -           | -           | -   |
| SVM     | $3.4e-05$   | $0.00037$   | $4.0e-12$   | -           | -           | -           | -   |
| xGBoost | $5.9e-14$   | $0.97584$   | $< 2e-16$   | $5.1e-06$   | -           | -           | -   |
| GLMNet  | $0.03081$   | $5.1e-08$   | $2.9e-07$   | $0.62723$   | $1.6e-10$   | -           | -   |
| kNN     | $1.3e-08$   | $< 2e-16$   | $0.00541$   | $5.8e-14$   | $< 2e-16$   | $7.2e-14$   | -   |

**Computation time on test data**

- *Global Friedman Test* Friedman rank sum test [33] rejects the global null hypothesis of no differences (p-value < 2.2e-16). This means we can conduct pairwise post hoc tests ($\alpha = 0.05$) with the null hypothesis of no difference.
- *Post Hoc Nemenyi Test* [63], see Table 3.

Table 3: Pairwise comparisons of algorithm performance with the Nemenyi test based on computing time on testing data. Underlined values indicate differences significant at $\alpha = 0.05$ level.

|         | LR          | RF          | CART        | SVM         | xGBoost     | GLMNet      | kNN |
|---------|-------------|-------------|-------------|-------------|-------------|-------------|-----|
| RF      | $< 2e-16$   | -           | -           | -           | -           | -           | -   |
| CART    | $0.676$     | $< 2e-16$   | -           | -           | -           | -           | -   |
| SVM     | $0.652$     | $6.5e-14$   | $0.019$     | -           | -           | -           | -   |
| xGBoost | $< 2e-16$   | $0.996$     | $< 2e-16$   | $7.2e-14$   | -           | -           | -   |
| GLMNet  | $3.2e-09$   | $6.2e-06$   | $1.2e-13$   | $4.0e-05$   | $1.9e-07$   | -           | -   |
| kNN     | $9.1e-14$   | $0.177$     | $6.8e-14$   | $2.3e-12$   | $0.034$     | $0.106$     | -   |

Table 4 provides the mean results of the classifier comparisons. (Recall that for train/test time: the lower, the better)

Table 4: Mean results of the classifier comparisons.

|            | LR    | RF    | CART  | SVM    | xGBoost | GLMNet | kNN   |
|------------|-------|-------|-------|--------|---------|--------|-------|
| Accuracy   | 0.761 | 0.854 | 0.831 | 0.8113 | 0.820   | 0.763  | 0.789 |
| Train Time | 0.370 | 7.019 | 0.199 | 1.866  | 9.561   | 1.491  | 0.012 |
| Test Time  | 0.062 | 0.458 | 0.055 | 0.106  | 0.407   | 0.184  | 0.291 |

As becomes evident from the mean values of the three quality metrics and the single-criterion test results presented above, there is no classifier that is significantly dominated by another classifer w.r.t. all three quality metrics. Hence, the marginal-front would contain all classifiers and would be rather indiscriminative compared to the empirical GSD-front that we present in the paper, see Figure 2 Appendix C.1, which contains random forest (RF), trees (CART), and k-nearest neighbor (kNN). This is in line with our explanation of OpenML results above. Since the quality metrics accuracy, train time, and test time are only weakly (if at all) correlated due to a trade-off between speed and accuracy, the marginal-front based on single-criterion comparisons does not facilitate practitioners' decision-making, while our empirical GSD-front provides valuable insights.

For the sake of completeness, we also report the results of these multiple single-objective evaluations on the PMLB benchmark suite in tables 5, 6, 7, and 8. The interpretation is completely analogous

Table 5: Post Hoc Nemenyi Test (Accuracy) on PMLB.

|           | cre     | svmRadial | J48     | ranger  | knn     | glmnet |
|-----------|---------|-----------|---------|---------|---------|--------|
| svmRadial | 0.74628 | -         | -       | -       | -       | -      |
| J48       | 0.78740 | 0.08257   | -       | -       | -       | -      |
| ranger    | 0.00106 | 0.09912   | 2.2e-06 | -       | -       | -      |
| knn       | 0.00227 | 4.2e-06   | 0.13239 | 2.0e-13 | -       | -      |
| glmnet    | 1.00000 | 0.67200   | 0.84844 | 0.00064 | 0.00360 | -      |

Table 6: Post Hoc Nemenyi Test Summary (Accuracy with Noisy X) on PMLB.

|           | cre    | svmRadial | J48    | ranger  | knn    | glmnet |
|-----------|--------|-----------|--------|---------|--------|--------|
| svmRadial | 0.8130 | -         | -      | -       | -      | -      |
| J48       | 0.4971 | 0.0323    | -      | -       | -      | -      |
| ranger    | 0.3063 | 0.9647    | 0.0019 | -       | -      | -      |
| knn       | 0.0072 | 3.8e-05   | 0.5290 | 5.1e-07 | -      | -      |
| glmnet    | 0.7173 | 0.0826    | 0.9994 | 0.0067  | 0.3195 | -      |

to the interpretation of the results on OpenML above. Note that the Friedman rank sum test rejects global null of no differences for all three criteria. This means we can conduct pairwise post hoc tests ($\alpha = 0.05$) with (two-sided) null of no difference.

### C.1.4 Discussion of the unexpected results

Recall the discussion in Section 5.1 about the unexpected results. We want to emphasize that these have a high degree of originality and should be of particular interest to practitioners. This shows that experience and intuition with a method can also be misleading if only the evaluation framework is slightly modified: A multidimensional quality metric that seeks the optimal trade-off between different, potentially conflicting metrics will generally rank differently than a unidimensional one. In the following, we show that the dominance of CART over xGBoost, SVM, LR, and GLMNet is indeed consistent with the quality metrics provided by the OpenML repository.

First of all, here, we are interested in the trade-off between accuracy and computation time, (e.g., the better the accuracy, the higher/worse the computation time). We now look at the comparison between SVM and CART to demonstrate that the results are indeed in line with the data. We obtain:

- For 27 datasets, CART outperforms SVM on all dimensions (e.g., prediction accuracy, computation time on test data, and computation time on training data) at once.

- For 9 datasets, CART dominates SVM for at least one quality metric and for all other quality metrics the performance of CART is not worse.

- For 41 datasets, SVM's prediction accuracy is better than CART's. At the same time, CART outperforms SVM for at least one of the two computation times. The two classifiers are therefore incomparable for these datasets.

- For 3 datasets CART outperforms SVM based on accuracy, but at least one of the computation times of SVM is below the one of CART.

Overall, there exists no dataset where SVM dominates CART in all dimensions at once. Either the two classifiers are incomparable, or CART dominates SVM. Furthermore, CART dominates SVM

Table 7: Post Hoc Nemenyi Test Summary (Accuracy with Noisy Y) on PMLB.

|           | cre     | svmRadial | J48     | ranger  | knn     | glmnet |
|-----------|---------|-----------|---------|---------|---------|--------|
| svmRadial | 1.00000 | -         | -       | -       | -       | -      |
| J48       | 0.03722 | 0.04911   | -       | -       | -       | -      |
| ranger    | 0.06405 | 0.04911   | 1.7e-07 | -       | -       | -      |
| knn       | 0.00096 | 0.00141   | 0.90728 | 2.3e-10 | -       | -      |
| glmnet    | 0.67200 | 0.73193   | 0.68732 | 0.00031 | 0.12513 | -      |

Table 8: Mean Results (Accuracy and Noisy Data) on PMLB

|  | cre | svmRadial | J48 | ranger | knn | glmnet |
|---|---|---|---|---|---|---|
| Accuracy | 0.7807 | 0.8494 | 0.8347 | 0.8629 | 0.7780 | 0.8106 |
| Accuracy with noisy x | 0.7307 | 0.7823 | 0.7679 | 0.7924 | 0.7339 | 0.7570 |
| Accuracy with noisy y | 0.7346 | 0.7776 | 0.7638 | 0.7984 | 0.7237 | 0.7640 |

for nearly half of the datasets (27 + 9 = 36 of 80). Thus, the dominance structure provided by our method is in line with the performance evaluation values provided by OpenML.

A second issue that may have influenced the unexpected performance structure obtained in the paper is the way performance is evaluated by OpenML. OpenML is based on the uploads of its users. Each user is free to decide which hyperparameter settings to use. Thus, as there might be a different goal on the hyperparameter setting in each dataset, the results are not representative for the best performance of each algorithm. This aspect should be included in any further discussion. Especially since some algorithms are more dependent on hyperparameter settings/tuning than others. For an example involving hyperparameter tuning that is fixed in advance, see Section 5.2.

## C.2 Experiments with PMLB

This sections give further insight to the exemplary benchmarking analysis on the Penn Machine Learning Benchmarks (PMLB) in Section 5.2 in the main paper. We start by giving more details on the data sets with all the computation settings of the classifier algorithms. Afterwards, we provide more figures and explanations of the analysis.

### C.2.1 Data

Penn Machine Learning Benchmarks (PMLB) is a collection of curated benchmark datasets for evaluating and comparing supervised machine learning algorithms [64]. We select all datasets from PMLB for binary classification tasks with 40 to 1000 observations[14] and less than 100 features. On these 62 datasets[15], a recently proposed classifier based on compressed rule ensemble learning [62] is compared w.r.t. robust accuracy against five well-established classifiers, namely classification tree (CART), random forest (RF), support vector machine with radial kernel (SVM), k-nearest neighbour (kNN), and generalized linear model with elastic net (GLMNet). In detail, we deploy

- *Support Vector Machine* (SVM) algorithm as implemented in the `e1071` library [26]
- *Random Forests* (RF) algorithm as implemented in the `ranger` library [88], requiring `e1071` library [26] and `dplyr` [87]
- *Decision Tree* (CART) algorithm (C4.5-like trees) as implemented in the `RWeka` library [39],
- *Elastic net* (GLMNet) algorithm is implemented through the `glmnet` library [32] requiring library `Matrix` [7], and
- *k-nearest neighbors* (kNN) algorithm as implemented in the `kknn` library [38].

Note that we used the respective methods in the `caret` library [49] for hyperparameter tuning and cross-validation to retrieve i) through iii), as detailed below.

We operationalize the latent quality criterion of robust accuracy through i) classical accuracy (metric), ii) robustness w.r.t. noisy features (ordinal), and iii) robustness w.r.t. noisy classes (ordinal). Computation of i) is straightforward; in order to retrieve ii) and iii), we follow [92, 93] by randomly perturbing a share (here: 20 %) of the data points. We randomly selected data points with a selection probability of $20\%$ and replaced the values by a random draw from the marginal distribution of the corresponding variable. (This is a slight difference to [92, 93] who replaced the data points by a random draw from a uniform distribution of the corresponding support of the marginal distribution.)

We then tune the six classifiers' hyperparameters on a (multivariate) grid of size 10 following [49] for each of the 62 datasets and eventually compute i) to iii) through 10-fold cross validation.

---

[14][49] requires at least 4 data points in the test set, which translates to a mininmal $n$ of 40, since we deploy 10-fold cross validation.

[15]Last access of PMLB: 12/05/24.

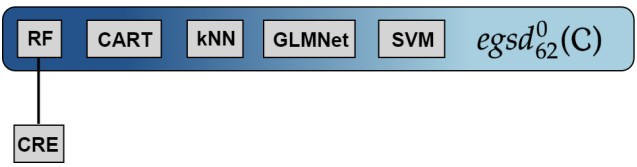

Figure 4: Hasse graph of the empirical GSD-relation for the PMLB data sets. The blue shaded region symbolizes the 0-empirical GSD-front, see Definition 6 ii).

### C.2.2 Detailed results of the GSD-based analysis

To initially obtain a purely descriptive overview, we construct the Hasse graph illustrating the empirical GSD relation. In this process, we calculate the value $d_{62}(C, C')$ for $C \neq C' \in \mathcal{C} :=$ {CRE, SVM, RF, CART, GLMNet, kNN} and connect $C$ to $C'$ with a top-down edge whenever $d_{62}(C, C') \geq 0$. The resulting graph is portrayed in Figure 4. It is evident from the graph that RF (strictly) empirically GSD-dominates the classifier CRE. All other classifiers are pairwise incomparable. Five classifiers, namely RF, CART, kNN, GLMNet, and SVM are not strictly empirically GSD-dominated by any other considered classifier and, thus, form the 0-empirical GSD-front.

This latter purely descriptive analysis already hints at the CRE not belonging to the GSD-Front. In order to transition to inferential statements, we aim to statistically test (at level $\alpha = 0.05$) whether CRE significantly lies in the GSD-front of some subset of $\mathcal{C}$. As detailed in Section 4.1, we conduct five pairwise tests for the hypothesis pairs $(H_0^{C'}, \neg H_0^{C'})$ (where $C := $ CRE and $C' \in \mathcal{C} \setminus$ {CRE}) at a level of $\frac{\alpha}{5}$, as explained in Section 4.[16] In other words, we test five auxiliary null hypotheses, each asserting that CRE is GSD-dominated by SVM, RF, CART, GLMNet, and kNN, respectively.

The results of these tests are visualized in Figure 5 (densities) and Figure 6 (cumulative distribution functions).[17] They indicate that the pairwise tests of CRE versus SVM, RF, CART, GLMNet, and kNN do not reject at a level of $\frac{\alpha}{5}$ nor at level $\alpha$. Hence, we conclude that based on the observed benchmark results we cannot conclude at significance level $\alpha = 0.05$ that CRE lies in the GSD-front of any subset of $\mathcal{C}$. In other words, we have no evidence to rule out that CRE is in the GSD-front, i.e., we cannot confirm based on the data that CRE is not outperformed by SVM, RF, CART, GLMNet, and kNN with respect to all compatible utility representation of robust accuracy. As can be seen in Figure 5, testing CRE vs. CART results in the smallest p-value of all pairwise tests, which appears plausible, since CRE is a CART-based method. On the other hand, the observed test statistic of CRE vs. RF is far away from the critical value and the test cleary does not reject, even though RF is also a tree-based method.

Finally, as discussed in Section 4.2, we further analyze the robustness of this test decision under contamination of the benchmark suite, i.e., deviations from the *i.i.d.*-assumption. As opposed to our OpenML analysis in Section 5.1, see also Appendix C.1, contamination does not affect the test decisions here, since none of the tests rejects already for uncontaminated samples. Increasing contamination only drives $p$-values further. The results are visualized in Figure 7. It is observed that the tests are neither significant at a level of $\frac{0.05}{5}$ nor at $0.05$ and this clearly does not change with growing size of contaminated benchmark data sets.

In summary, the PMLB experiments demonstrated how to apply our benchmarking framework to the problem of comparing a newly proposed classifier to a set of state-of-the-art ones. Furthermore, it illustrated our tests' applications to multiple criteria of mixed scales (ordinal and cardinal) that operationalize a latent performance measure, namely robust accuracy. It became evident that our framework allows to statistically assess whether the novel classifier CRE can compete with existing ones - that is, whether CRE lies in the GSD-front of some state-of-the-art classification algorithms. In

---

[16]As clarified in Footnote 7, the tests in Sections 5.1 and 5.2 are based on the unregularized test statistics $d_s^0(C', C)$.

[17]For generating these plots, we used quantile functions from both base r and the ggplot library. As the underlying quantile functions definitions differed slightly, we relied on the latter and corrected the quantiles from the other manually. Detailed documentation of all computations involved in generating the visualizations in the paper, we refer the interested reader to `https://github.com/hannahblo/Statistical-Multicriteria-Benchmarking-via-the-GSD-Front`.

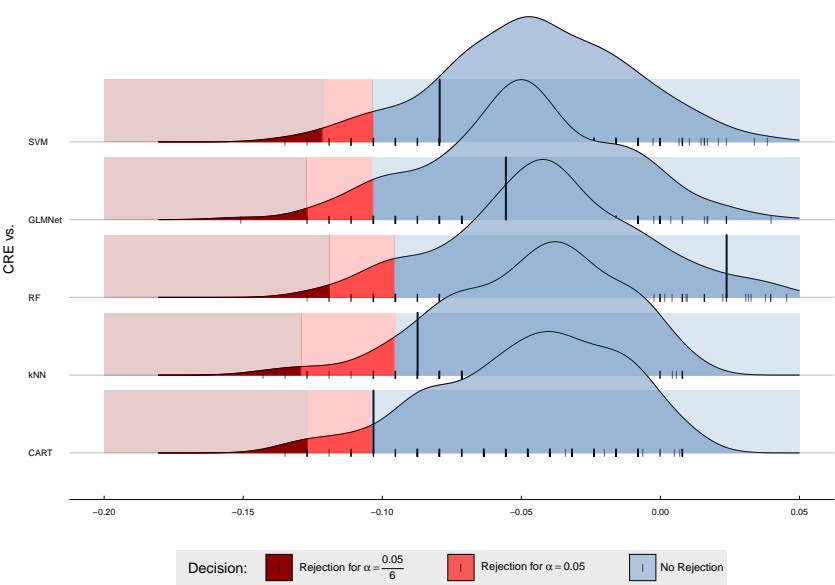

Figure 5: Densities of resampled test statistics for pairwise tests of CRE vs. six other classifiers on 62 datasets from PMLB. Big (small) vertical lines depict observed (resampled) test statistics. Rejection regions for the static (dynamic) GSD-test are highlighted red (dark red). As becomes evident, we cannot reject any of the pairwise tests for neither significance level.

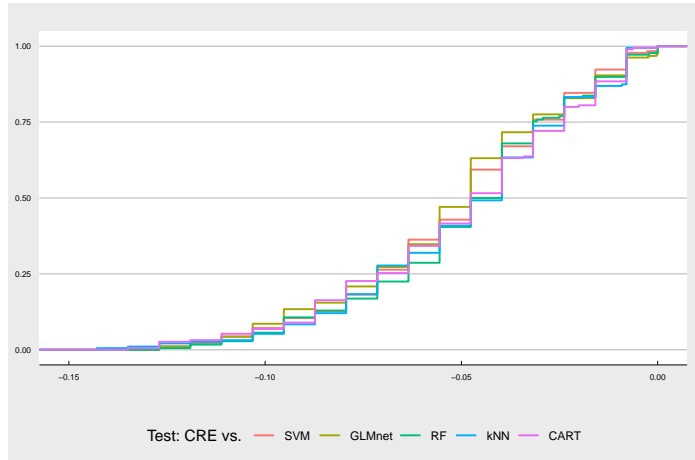

Figure 6: Cumulative Distribution Functions (CDFs) of resampled test statistics for hypothesis tests on PMLB benchmark suite of CRE vs. SVM, GLMNet, RF, kNN, and CART, respectively. As opposed to Figure 5 above, values of observed test statistics are not included. They are: $-0.1031746$ (CART), $-0.08730159$ (kNN), $0.02380952$ (RF), $-0.05555556$ (GLMNet), $-0.07936508$ (SVM). It becomes evident that the resampled test statistics' distributions are more similar to each other than in the case of testing SVM vs. competitors in the OpenML benchmark suite.

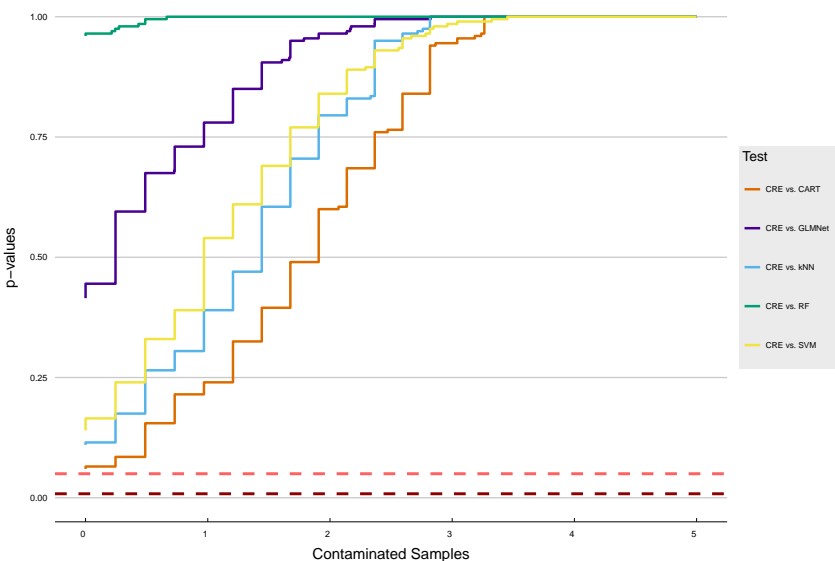

Figure 7: Effect of Contamination: $p$-values for pairwise tests of CRE versus the five competitors in PMLB benchmark suite application. Analogous to Figure 5, dotted red lines mark significance levels of $\alpha = 0.05$ (dark red: $\alpha = \frac{0.05}{6}$). Since none of the tests reject for $\alpha = 0.05$ under no contamination, this obviously does not change with contaminated samples.

this case, the test decisions of both static and dynamic GSD-tests was not to reject the null hypothesis of CRE being outperformed by RF, CART, SVM, GLMNet, and kNN w.r.t. to robust accuracy.

