# OpenReview forum: "Statistical Multicriteria Benchmarking via the GSD-Front"
_NeurIPS.cc/2024/Conference — NeurIPS 2024 spotlight_

### Official Review · Reviewer_n7Tx · 2024-07-06

**Soundness:** 3
**Presentation:** 3
**Contribution:** 3
**Rating:** 7
**Confidence:** 4

**Summary:**

The authors propose a novel way of comparing classifiers to assess their effectiveness. They posit (1) that comparisons should allow for different quality metrics simultaneously, (2) that comparisons should take into account the statistical uncertainty induced by the choice of benchmark suite, and (3) that the robustness of the comparisons under small deviations in the underlying assumptions should be verifiable. They illustrate their proposed method on the benchmark suite PMLB and on the platform OpenML.

**Strengths:**

The paper is clear and easy to follow. The proposed method is innovative, it enjoys some desirable mathematical properties (that are well-presented), and it seems to work well on the selected benchmarks.

**Weaknesses:**

Some references to the existing imprecise probabilistic literature are missing, there are some small typos, and an extra effort can be put to make the definition of $d_s$ more accessible.

**Questions:**

When introducing imprecise probabilities, the authors fail to reference a few papers in such a literature. In particular, the authors leverage the idea of $\epsilon$-contamination in section 4.2. Other papers in the imprecise probabilistic machine learning literature dealing with, and advancing the knowledge of, $\epsilon$-contaminations are [1-5]. In addition, [6] uses IPs to robustify Bayesian neural networks. More in general, we suggest the authors to read (and possibly cite) the works by Eyke Hüllermeier, Thierry Denoeux, Sebastien Destercke, Fabio Cuzzolin, Alessio Benavoli, and Michele Caprio on imprecise probabilistic machine learning, all of which have a distinct "robustness" flavor to them, and that might be interesting to the authors for future endeavors.

There is a typo in line 95: there's a period after superscript 3.

While I understand the need of being as general as possible, wouldn't it be more beneficial to the reader to define $\hat{\pi}_c(\{z\})= 1/s \left|\lbrace\{ i : i \leq s \text{, } \Phi(C,T_i)=z \rbrace\}\right|$ in line 167, instead of using a generic $M \subseteq [0,1]^n$ for the definition?

There is a typo in line 167: last letter should be $C^\prime$ and not $C$.

Line 205: the question *of* how to [...]

---

[1] https://www.jstor.org/stable/2242055

[2] https://onlinelibrary.wiley.com/doi/book/10.1002/9780470434697 (Chapter 10)

[3] https://link.springer.com/chapter/10.1007/978-3-031-57963-9_1#:~:text=In%20their%20seminal%201990%20paper,bound%20to%20hold%20with%20equality.

[4] https://arxiv.org/abs/2402.00957

[5] https://arxiv.org/abs/2308.14815

[6] https://arxiv.org/abs/2302.09656

**Limitations:**

See questions.

---

> ### Author Rebuttal · Authors · 2024-08-06
>
> First of all, we would like to thank the reviewer for their time and thoughtful consideration of our paper. We believe that the more direct mathematical notation suggested by the reviewer regarding the empirical measure as well as the suggested more detailed explanation of the test statistic will help to further improve the accessibility of the paper, which the reviewer otherwise describes as “easy to follow”.
>
> We are pleased that the reviewer finds our method “innovative” and emphasizes that it “enjoys some desirable mathematical properties”, which they also describe as “well-presented”.
>
> We will now address the reviewer's concerns and questions in turn:
>
> **Weakness 1: Missing references, different notions of robustness in IP for ML**
>
> We agree with the reviewer that Huber's book on robust statistics should definitely be cited when introducing the contamination model and thank the reviewer for identifying this gap. We add a corresponding reference to the revision of our paper.
> We are also very grateful for the further references provided by the reviewer; they for sure deserve proper consideration: We will go through them and use some part of the additional page to add appropriate references in the revision of our related work section and the introduction of contamination models in Section 4.2.
> In this context, we will also pay special attention to sharpen the presentation of our understanding of robustness even more, also in contrast to related, but in detail subtly different concepts from current literature.
> Thanks once more for raising the issue of slightly deviating robustness concepts and for all the hints to important further references.
>
>
> **Weakness 2: Extra effort to make the definition of $d_s$ more accessible**
>
> We agree with the reviewer that a purely formal introduction of the test statistic $d_s$, which plays a central role in the paper, is not ideal. We are happy to use some of the additional space in the revised version of the manuscript to add an additional – less formal and instead more intuitive – description of $d_s$. We think this is an excellent point and thank the reviewer for raising it.
>
> **Questions:**
>
> First of all, thank you for your close reading, which made it possible to find three notational and grammatical typos scattered throughout the paper. We appreciate this very much. We agree that your proposed definition of the empirical measure is more direct and more easy to access. We will change this accordingly in the revised version. Thank you very much for pointing this out!
>
> We thank you once more for your review and your concrete suggestions and questions allowing us to further improve our paper.

---

### Official Review · Reviewer_3sEe · 2024-07-09

**Soundness:** 4
**Presentation:** 3
**Contribution:** 3
**Rating:** 8
**Confidence:** 4

**Summary:**

The authors propose a method for multicriteria evaluation of classifiers which is more informative than the Pareto front. The new GSD-front and is based on the previously proposed generalized stochastic dominance ordering (GSD) for classifiers. The authors also provided a sound inference framework, included hypothesis tests for whether a new classifier would be in the GSD-front of a benchmark.

**Strengths:**

1. The method is sound and extremely useful
2. The paper is very well written; notation is consistent throughout
3. Given the large (and increasing) number of potential models that can be applied to any given task, this method is very significant for identifying which models tended to be among the best for other similar tasks
4. The hypothesis tests derived from the framework are also very useful to add significance to these benchmarks
5. All the information needed to fully understand the paper is in the main body of text

**Weaknesses:**

1. There could have been maybe a toy example to show the differences between GSD-front and Pareto-front in a more didactic way
2. The method seems computationally expensive. The pmlb_permutation_tests.R script needs 5 days to run and this does not include training of the evaluated models, as these are collected results

**Questions:**

1. How does the running time of a Pareto-front benchmark compare with a GSD-front one?
    1. If the Pareto-front is much cheaper to obtain, would it make sense to start with it, and only look for what classifiers in that pre-selection would be in the GSD-front?
2. How were different hyper-parameter values treated in the experiments? Were all results of each classifier for a dataset aggregated or did the authors use specific settings for each classifier, e.g. default values from the classifier's implementation?

**Limitations:**

Yes

---

> ### Author Rebuttal · Authors · 2024-08-06
>
> First of all, we would like to thank the reviewer for their time and thoughtful consideration of our paper. We are grateful for the suggestion to add a toy example showing the differences between GSD-front and Pareto-front in a more didactic way and the suggestion to add more details on hyperparameter tuning in the PMLB experiment. We follow both suggestions in the revision of our manuscript (the example will be added to the main text, the tuning details to the corresponding paragraph in the appendix).
>
> We are very pleased that the reviewer appreciates our method as “extremely useful” and the paper as “very well written”, with notation being “consistent throughout”.
> We will now address the reviewer's comments in turn:
>
>
> **Weakness 1: Missing toy example**
>
> We agree that a concrete (minimal) example could help to quickly and intuitively grasp the differences between Pareto and GSD-Front. We are happy to use parts of the additional page to include such an example. Specifically, we plan to build the example directly after Definition 1 to illustrate the Pareto front and then continue it after Definition 6 to contrast the GSD front. To further strengthen the reader's intuition right from the start, we also plan to refer to the "Method comparison" from Section 5.1 at the end of these examples. Thanks for the proposal, we think this is an excellent idea!
>
>
> **Weakness 2: Computation time**
>
> We agree that the computation time is relatively long. However, we believe that this is not really a major problem for the settings in which our approach is intended to be applied. Firstly, we are in a benchmark setup, so our method only needs to be run once, not several times. This is in contrast to the intensive hyperparameter tuning of models that is often part of everyday work. Even more important, our method can provide the information that an algorithm is dominated by others and, therefore, should not be considered in later tasks. So, starting with a rather expensive benchmark experiment can save a lot of time later on, in particular, as the results provided by our method come with (additionally robustified) inferential guarantees and are not purely descriptive.
>
> Secondly, the expensive part is the computation of the test (as it is a permutation test) and not the computation of the empirical GSD front. The computation of the test is indeed expensive, but unlike other benchmarking approaches, we obtain from it a statistically sound inferential statement. If we restrict our analysis to the empirical, non-inferential part, the computation time is much smaller.
>
>
> **Question 1: Pareto-front vs empirical GSD-front**
>
> We think the idea to first compute the Pareto-front (as a kind of pre-processing step) and then proceed with our GSD-based analysis is excellent. Thank you very much for that suggestion! We will include this idea in the new paragraph with end-user recommendations (see the answer to 1fNM) in the revised manuscript.
> Concerning the computation of the Pareto vs GSD-front: Since the Pareto-front simply checks whether one classifier is strictly dominated by another, the Pareto-front is simply based on counting the existence of observed dominances. This can be done in milliseconds. As we pointed out above, the computation of the empirical GSD-front is much faster than the entire inference part, but still more expensive than the computation of the Pareto-front. Similar to above, we think that the further insight that the empirical GSD-front provides more than justifies the higher computational time – in particular, as it can save computation time later on.
>
>
> **Question 2: Hyper-parameter**
>
> Thanks for this important question. In the case of the Open ML platform, we used the performance evaluation provided by the OpenML library, see line 838-846. In the case of the PMLB benchmarking suite, we tuned the six classifiers’ hyperparameters on a (multivariate) grid for each of the 62 datasets and eventually computed all evaluation metrics through 10-fold cross validation, see lines 999-1008 in the paper for more details. Concretely, we tuned
>
> - “sigma” and “C” for support vector machines
> - “mtry”, “splitrule” and “min.node.size” for ranger
> - “k” for knn
> - “alpha” and “lamba” for glmnet
> - “CF”, “M”, and “U” for CART (J48)
> - “eta” and “k” for compressed rule ensembles (cre)
>
>
>
> We thank you once more for your review and your concrete suggestions and questions allowing us to further improve our paper.

---

### Official Review · Reviewer_1fNM · 2024-07-12

**Soundness:** 3
**Presentation:** 3
**Contribution:** 3
**Rating:** 7
**Confidence:** 3

**Summary:**

This submission studies the problem of comparing multiple classifiers under multiple evaluation criteria. It presents the construction of the GSD-front, i.e., the set of GSD-undominated classifiers, and an empirical variant called the $\epsilon$-empirical GSD-front.

Then, theoretical aspects of the GSD-front and the $\epsilon$-empirical GSD-front, such as satistical consistency, and the inclusion concerning the respective Pareto-front, are investigated.

Two statistical tests, namely static GSD-test and dynamic static GSD-test, on whether a given classifier $C \in \mathcal{C}$ is an element of the true GSD-front are proposed.  Both tests are shown to be valid level-$\alpha$ tests. A robustified static GSD-test is also proposed to deal with the setting where the available data sets are drawn under non-i.i.d.-scenarios.

An empirical study is conducted to assess the proposed tests. It covers 80 binary classification data sets from OpenML [70]. The set of classifiers consists of Support Vector Machine (SVM) with Random Forest (RF), Decision Tree (CART), Logistic Regression (LR), Generalized Linear Model with Elastic net (GLMNet), Extreme Gradient Boosting (xGBoost), and k-Nearest Neighbors (kNN). The set of evaluation metrics consists of predictive accuracy, computation time on the test data, and computation time on the training data. Empirical evidence suggests that the GSD-approach tends to be more contrastive, compared to Pareto-front and another test that aggregates the single-metric tests as in [19].

Another empirical study is conducted using 62 datasets from the Penn Machine Learning Benchmark (PMLB). The set of classifiers consists of rule ensembles of trees (CRE) [53],  CART, RF, SVM with radial kernel, kNN and GLMNet.  The set of evaluation metrics consists of classical accuracy (metric), ii) robustness of accuracy w.r.t. noisy features (ordinal), and iii) robustness of accuracy w.r.t. noisy classes (ordinal). The results provided by GSD-tests are discussed.

**Strengths:**

S1: The submission is well-written and organized.

S2: Theoretical aspects of the GSD-front and GSD-tests are carefully investigated.

S3: The authors also take into account an interesting aspect that the different evaluation metrics may have different levels of impact. Non-i.i.d.-scenarios are also taken into account.

S4: Empirical evidence seems to suggest that GSD-tests can complement the existing statistical tests for comparing classifiers meaningfully.

**Weaknesses:**

W1: The authors might consider mentioning more scenarios where comparing multiple classifiers under the multiple evaluation criteria is crucial. An example might be multi-label classification, where multiple evaluation metrics have been proposed to evaluate multi-label classifiers. The set of evaluation metrics can also be extended with others, such as training time, storage memory, and so on.

W2: The motivation to compare multiple classifiers using evidence on multiple data sets, which can come from very different application domains might need to be strengthened. For example, it would be not easy to convince practitioners in safety-critical applications, such as lung cancer detection to use some classifier that has shown to be promising given evidence aggregated from multiple domains.

W3: Additional recommendations for the end-user might be beneficial. For example, what should one do if the GSD-front contains multiple classifiers?

**Questions:**

Q1: Could you elaborate more on the case where only one training data is taken into account? This is related to the comment in W2.

Q2: Regarding W3, should one randomly choose one element of the GSD-front when having to make predictions? Yet, one might consider doing ensemble learning on the set of promising classifiers.  In safety-critical applications, such a strategy might lower the interpretability and explainability of the predictive system. Do you have any recommendations in such cases?

Q3: Could you also discuss the Pareto-front and results of the test that aggregates the single-metric tests as in [19] for the empirical study on PMLB data sets?

**Limitations:**

Please refer to "Weaknesses" and "Questions" for detailed comments.

---

> ### Author Rebuttal · Authors · 2024-08-06
>
> Thank you for your time and thoughtful consideration. We appreciate the suggestions to include end-user recommendations and comparative studies for the PMLB datasets. We follow both in the revision (recommendations go to the main paper, studies to the appendix). We are pleased you find our paper "well-written and organized" and our framework "carefully investigated".
>
> We now address your comments:
>
> **W1:** We are grateful for your idea to add more scenarios, which we do in the revised version. The multilabel classification case is very interesting: For classes $i=1,...,K$, one can use a collection of classical metrics for multiclass classification like Hamming distance, Jaccard similarity etc. Within our approach, a more direct way may be even more interesting: Take a separate metric for every class (e.g., accuracy) and then treat all metrics together as a K-dimensional multi-cardinal metric. With the flexibility of the $R_2$ relation, one can implement different strengths of the underlying scale of measurement, e.g. by imposing only cardinal commensurability within each class, or by additionally requiring interclass commensurability (which presumably leads to an approach very similar to using only a classical (cardinal) one-dimensional metric for the entire multiclass performance). This can be accompanied, similar to our experiments, with further criteria such as training time or memory.
>
> **W2:** We fully share your concern about benchmarking on datasets from multiple domains different from the one of intended use, e.g., lung cancer detection. We consider this a very relevant problem in benchmarking generally (not only limited to our GSD-analysis). It is the very reason why we robustified our tests against non-i.i.d. selection of datasets in section 4.2. We observe it to be common practice in benchmarking to simply test methods on huge benchmark suites that include datasets from a myriad of domains. Statistically, such a collection can hardly be considered *i.i.d.* from the population of interest.
> Our robustification allows for valid conclusions even if some heterogeneity/asymmetry in the selection processes is allowed, i.e., if it deviates from *i.i.d*. E.g., for the OpenML study, the observed conclusions remain valid under contamination (i.e. can come from very different data domains) of 7, 8, 11 and 11 out of 80 datasets respectively (lines 335-344). This exactly addresses your concern: Our methodology allows for valid statistical conclusions even if the datasets are not *i.i.d*.
> Finally, from a different angle, this point is an excellent motivation to extend our results to stratified and regression-like situations, as mentioned in Section 6. Clearly, the area from which a dataset originates is an important meta-information to include.
>
> **Q1:** Our method can also be applied to one dataset. It then translates to the partial order resulting from the classifiers' performances w.r.t. a multidimensional metric on this dataset, and, in the further special case of only one metric, even results in a total order. This is a sensible thing to do if one is interested in a purely descriptive (as opposed to inferential) analysis of classifier performance on a single dataset. Then, however, much of our conceptual work regarding statistical properties of our estimator and tests is not needed; the tiny sample size does not allow for meaningful inferences. Nevertheless, such an application of our methodology is possible in principle and leads to the known special cases.
>
> **Q2 and W3:** We agree it is good to add a paragraph with end-user recommendations. We use part of the extra page for such an explanation and will add it to the beginning of Section 5. Specifically, we plan to explain that our method is not primarily intended to identify the best classifier for a problem class. (In this sense, it is not a problem if the GSD front contains more than one element.) It is rather for checking whether a newly proposed classifier for a certain problem class can potentially improve on the state-of-the-art classifiers, or whether it disqualifies itself from the outset. Furthermore, we want to emphasize that our framework allows us to provide such statements with inferential guarantees by means of appropriately constructed statistical tests, and even to examine these inferential guarantees in terms of their robustness to deviations from the i.i.d assumption.
>
> We are very grateful for your idea about ensemble learning with the GSD-front. Although our original motivation was different, this sounds practically quite rewarding, but of course needs more careful attention than can be paid in the context of this paper under review. Speculating a bit, it seems tempting to apply aggregation rules to the outputs of the different classifiers in the GSD front. Most in line with our work is to rely then on a credal point of view, collecting, for every unit, all the predicted classes (similar to E-admissibility in credal decision making). In safety-critical applications, this may help to avoid overoptimism, and it informs the user about whether there is consensus among the non-dominated classifiers.
>
> **Q3:** We add the results of the Pareto-analysis and the aggregated single-metric tests for the PMLB study to the appendix. Also for the PMLB case, the Pareto front contains all classifiers and is not informative. The pdf page presenting the results of the single-metric tests as in [19] is attached to the “global response” above. Furthermore, note that – for all three metrics considered – the Friedman rank sum test rejects the global null of no differences such that we can conduct (two-sided) pairwise post hoc tests (α=0.05) with no difference as their null. The interpretation of the results is in complete analogy to the one of the OpenML study in Ap. C.1.3.
>
> We thank you once more for your review and your concrete suggestions and questions allowing us to further improve our paper.

---

> > ### Comment · Reviewer_1fNM · 2024-08-09
> >
> > Thanks for your reply. I think the response indeed complements the submission. The additional discussion on the case where GSD-front contains multiple classifiers might need to be further elaborated as it might be of practical relevance. In different applications of credal decision making, I guess one might expect that there are experts who know and can provide the true classes (by possibly paying additional costs). Yet, I think whether one should expect such experts in model selection/comparison might require more careful attention. However, I think the motivation for avoiding overoptimism is also interesting. After due consideration, I raised the rating to $7$.

---

> > > ### Author Response · Authors · 2024-08-09
> > > **Thanks for the reply**
> > >
> > > Thank you for your reply and for taking into account our rebuttal in your assessment of the paper!
> > >
> > > We will include a discussion about the case where the GSD front contains multiple classifiers in the new paragraph with end-user recommendations.

---

### Author Rebuttal · Authors · 2024-08-06

**Global response**

We sincerely thank all reviewers for their thorough, high-quality, and detailed assessment of our manuscript. We are encouraged by the very positive, affirmative reviews and feel most grateful for the very precise and constructive suggestions on how to improve our paper further. The reviews unanimously underscore the novel and innovative nature of our proposed benchmark methodology via the GSD-Front.


**Reviewer 1fNM** praises the paper for its "well-written and organized" presentation, emphasizing the thorough investigation into the theoretical aspects of the GSD-front and GSD-tests. The reviewer further welcomes the inclusion of different evaluation metrics and the consideration of non-i.i.d. scenarios as a crucial aspect of the methodology, noting that "the empirical evidence suggests that GSD-tests can complement the existing statistical tests for comparing classifiers meaningfully."

**Reviewer 3sEe** highlights the method's soundness and utility by calling it "extremely useful" and “very significant”. The reviewer further praises the notation as “consistent throughout”. Generally, 3sEe applauds the "very well written" nature of the paper, which provides "all the information needed to fully understand the paper in the main body of text."

Lastly, **Reviewer n7Tx** calls our work “innovative”, stating that the paper is "clear and easy to follow" with a methodology that "enjoys some desirable mathematical properties", which are “well presented”. The reviewer also appreciates the extensive empirical evidence in favor of our method, stating that it "seems to work well on the selected benchmarks.”

Generally, neither the theoretical soundness nor the practical relevance ("excellent" and “extremely useful”: Reviewer 3sEe) was questioned. The reviews rather focused on didactical aspects. As described in detail in the individual responses, we are most confident that we can properly address all issues in the camera ready version. The additional page will mainly be used to follow the reviewers’ suggestions to add some further didactical material illustrating and demonstrating our approach. In particular, we will include a paragraph with additional recommendations for the end-user (**Reviewer 1fNM**), add an example elaborating the differences between GSD-front and Pareto-front (**Reviewer 3sEe**), as well as extend the discussion of related work and additionally provide a more intuitive description of the central test statistic (**Reviewer n7Tx**).

**Attached pdf page**

Please note that the pdf page attached to this global comment is specific to Q3 of Reviewer 1fNM. The context of the contents of the page is explained in our corresponding answer.

---

### Decision · Program_Chairs · 2024-09-25

**Decision:**

Accept (spotlight)

**Comment:**

The authors propose a new method for benchmarking classifiers on multiple criteria using the GSD-front. The paper provides an in depth treatment with theoretical results and statistical tests, backed up by extensive empirical experiments. On the down side, the paper is lacking in providing the reader with an intuitive understanding of the GSD-front, which the authors have promised to address via a didactic example. There is also a focus on benchmarking using large collections of datasets, which could be useful for academic comparisons of classifiers, but less useful for practitioners who are interested in selecting good classifiers for a particular problem.  Overall, all reviewers recommended accepting the paper.